# Increases in cyclin A/Cdk activity and in PP2A-B55 inhibition by FAM122A are key mitosis-inducing events

Benjamin Lacroix [1,2,5], Suzanne Vigneron[1,2,5], Jean Claude Labbé[1,2], Lionel Pintard [2,3], Corinne Lionne [4], Gilles Labesse [4], Anna Castro [1,2✉] & Thierry Lorca [1,2✉]

## Abstract

Entry into mitosis has been classically attributed to the activation of a cyclin B/Cdk1 amplification loop via a partial pool of this kinase becoming active at the end of G2 phase. However, how this initial pool is activated is still unknown. Here we discovered a new role of the recently identified PP2A-B55 inhibitor FAM122A in triggering mitotic entry. Accordingly, depletion of the orthologue of FAM122A in *C. elegans* prevents entry into mitosis in germline stem cells. Moreover, data from *Xenopus* egg extracts strongly suggest that FAM122A-dependent inhibition of PP2A-B55 could be the initial event promoting mitotic entry. Inhibition of this phosphatase allows subsequent phosphorylation of early mitotic substrates by cyclin A/Cdk, resulting in full cyclin B/Cdk1 and Greatwall (Gwl) kinase activation. Subsequent to Greatwall activation, Arpp19/ENSA become phosphorylated and now compete with FAM122A, promoting its dissociation from PP2A-B55 and taking over its phosphatase inhibition role until the end of mitosis.

**Keywords** FAM122A; Arpp19; PP2A-B55; Mitosis; Cyclin A.
**Subject Categories** Cell Cycle; Post-translational Modifications & Proteolysis

## Introduction

Mitotic entry and progression are induced by massive protein phosphorylation resulting from the fine-tuned balance between the kinase cyclin B/Cdk1 and its counteracting phosphatase PP2A-B55 (Mochida et al, 2010; Gharbi-Ayachi et al, 2010; Mochida et al, 2009; Vigneron et al, 2009). Cyclin B/Cdk1 activity is maintained low during G2 by the Wee1/Myt1 kinases that phosphorylate Cdk1 on its inhibitory site tyrosine 15. At M phase entry, cyclin B/Cdk1 activity is triggered by a positive feed-back loop(Pomerening et al, 2003; Sha et al, 2003). A partial active pool of this kinase phosphorylates Wee1/Myt1, as well as the Cdc25 phosphatase responsible of Cdk1-tyrosine 15 dephosphorylation, promoting a rapid and complete activation of the cyclin B/Cdk1 complex. How cyclin B/Cdk1 partial activation is firstly triggered at G2-M is a main question yet to be answered. Inhibition of PP2A-B55 was proposed as the putative cause inducing partial cyclin B/Cdk1 pool activation and mitotic entry. PP2A-B55 is regulated by the Gwl-Arpp19/ENSA pathway (Mochida et al, 2010; Gharbi-Ayachi et al, 2010; Burgess et al, 2010; Hached et al, 2019). During G2, the activity of this phosphatase is high. Then, at mitotic entry, Gwl is activated and phosphorylates its substrates Arpp19/ENSA converting them into high affinity inhibitors of PP2A-B55. Decreased PP2A-B55 activity could then favour a partial Wee1/Myt1/Cdc25 phosphorylation and reactivation of cyclin B/Cdk1 triggering the feedback loop. However, Gwl activation itself depends on cyclin B/Cdk1 discarding the modulation of this phosphatase by the Gwl-Arpp19/ENSA pathway as the first event triggering cyclin B/Cdk1 firing and mitotic entry (Vigneron et al, 2011; Blake-Hodek et al, 2012). Moreover, although both Gwl and Arpp19/ENSA are essential for mitotic entry in *Xenopus* egg extract model (Gharbi-Ayachi et al, 2010; Vigneron et al, 2009), the depletion of these proteins do not prevent G2-M transition in human and mouse cells (Hached et al, 2019; Alvarez-Fernandez et al, 2013).

Besides cyclin B/Cdk1, cyclin A/Cdk is also present and is a poor substrate of Myt1 kinase during G2 (Coulonval et al, 2003; Booher et al, 1997). This kinase could participate to Wee1/Myt1/Cdc25 phosphorylation during G2-M if the activity of PP2A-B55 is partially inhibit during this transition. In this line, FAM122A has been recently described as a new inhibitor of PP2A-B55 that is present and active during G2.

FAM122A was firstly identified as an interactor of the PP2A-B55 complex that inhibits this phosphatase and promotes the degradation of PP2A catalytic subunit (Fan et al, 2016). This protein was additionally shown to negatively modulate PP2A-B55α in the nucleus. Its phosphorylation by Chk1 and its subsequent retention into the cytoplasm is essential for the G2/M checkpoint

[1]Université de Montpellier, Centre de Recherche en Biologie cellulaire de Montpellier (CRBM), CNRS UMR 5237, 1919 Route de Mende, 34293 Montpellier cedex 5, France. [2]Programme équipes Labellisées Ligue Contre le Cancer, Paris, France. [3]Université Paris Cité, Institut Jacques Monod, F-75013 Paris, France. [4]Centre de Biologie Structurale (CBS), CNRS, INSERM, Montpellier University, Montpellier, France. [5]These authors contributed equally: Benjamin Lacroix, Suzanne Vigneron. ✉E-mail: anna.castro@crbm.cnrs.fr; thierry.lorca@crbm.cnrs.fr

a

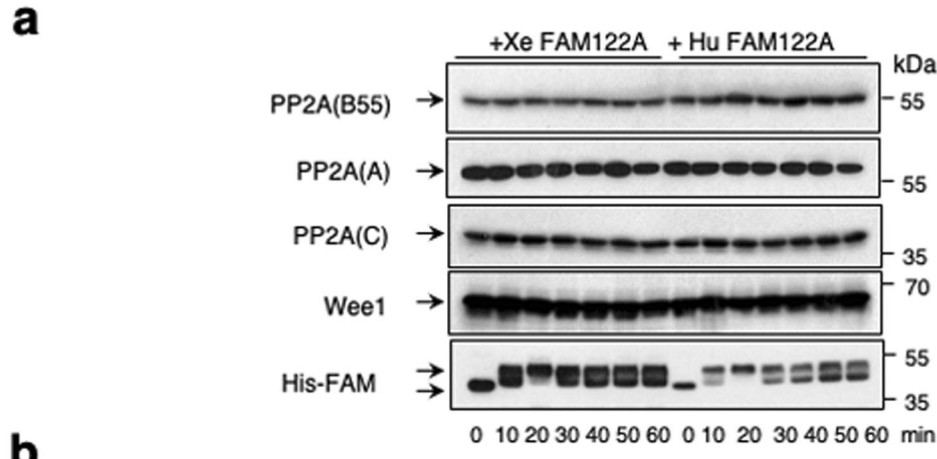

b

KINASE INACTIVATED EXTRACTS

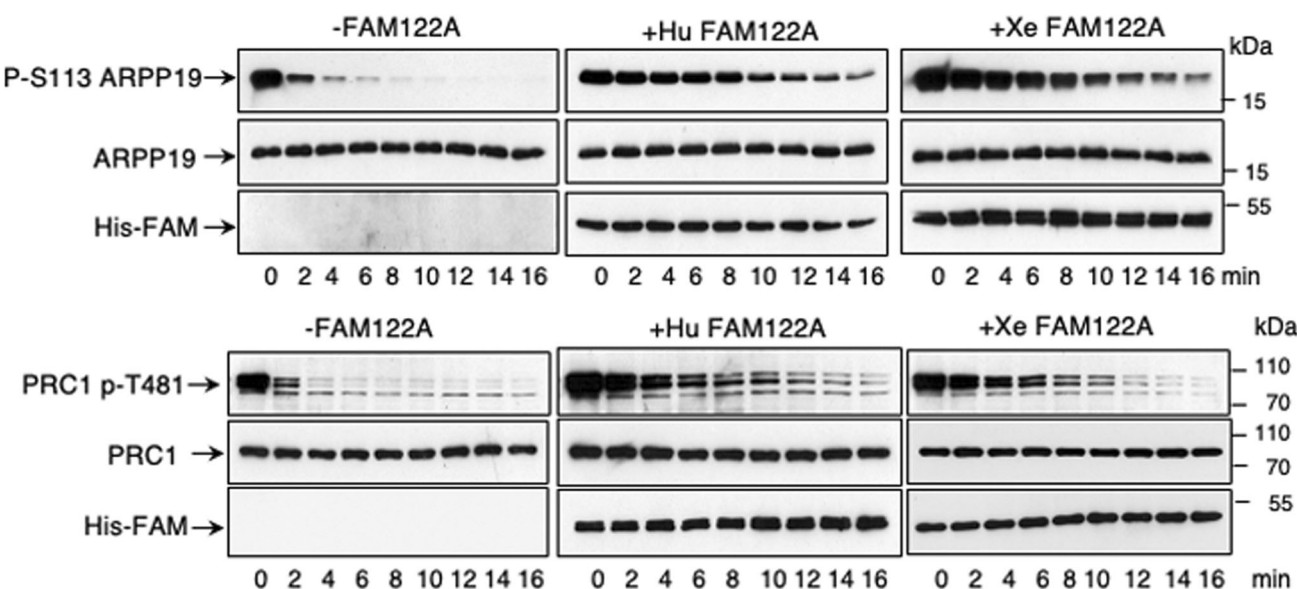

c

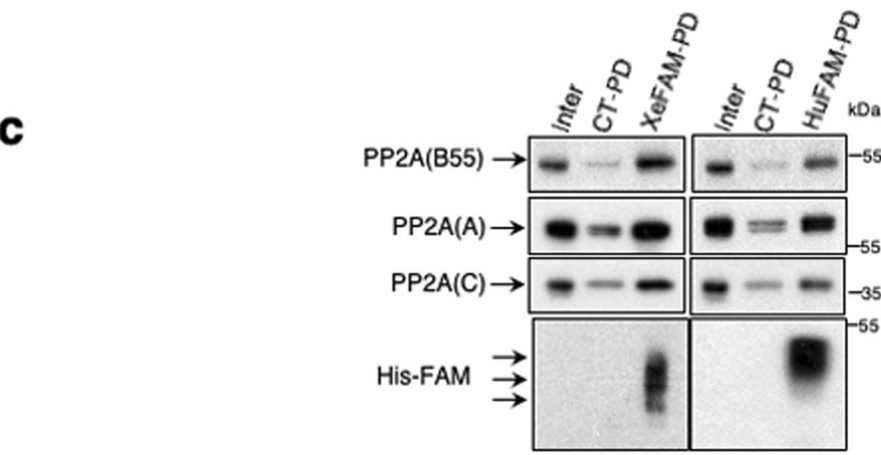

**Figure 1.   PP2A-B55 inhibition by FAM122A does not involve changes in PP2A-C or Wee1 levels.**

(A) 20 µl of interphase extracts were supplemented with a final concentration of 7.15 µM of *Xenopus* or of human FAM122A and 0,8 µl of the mix was recovered at different time points to measure the indicated proteins by western blot. (B) Arpp19 or PRC1 proteins "in vitro" phosphorylated by PKA or by cyclin A/Cdk1 respectively were supplemented at a final concentration of 1.65 µM together with *Xenopus* or human FAM122A (final concentration of 14.3 µM) to kinase inactivated interphase extracts and the dephosphorylation rate of S113 of Arpp19 and T481 of PRC1 as well as the levels of *Xenopus* and human FAM122A were analysed by western blot. (C) Interphase extracts were supplemented (Xe/HuFAM-PD) or not (CT-PD) with either *Xenopus* or human FAM122A as in (A) and used for histidine pull-down. The levels of PP2A B55, A and C subunits as well as the amount of FAM122A were checked by western blot. Source data are available online for this figure.

by preventing the nuclear inhibition of this phosphatase and permitting the dephosphorylation and stabilisation of Wee1 (Li et al, 2020). By its presence in G2 and its capacity to specifically inhibit PP2A-B55, FAM122A is a good candidate to fulfil the role of PP2A-B55 inhibitor able to potentiate cyclin A/Cdk-dependent phosphorylation at G2 and to trigger M phase. In this study we investigated the role of FAM122A in mitotic entry and we demonstrated that, by inhibiting PP2A-B55, this protein promotes a first burst of cyclin A/Cdk-dependent phosphorylation that will induce cyclin B/Cdk1 activation and mitotic entry.

# Results

## PP2A-B55 inhibition by FAM122A does not involve changes in PP2A-C or Wee1 levels

The role of FAM122A as a specific PP2A-B55 inhibitor has been reported by two different laboratories (Fan et al, 2016; Li et al, 2020). One laboratory reported a role of this protein in the negative modulation of PP2A-B55 via the degradation of the PP2A-B55 catalytic subunit C (Fan et al, 2016). The second one suggested that the inhibition of PP2A-B55 by FAM122A induces the dephosphorylation and degradation of Wee1. We checked these two premises in interphase *Xenopus* egg extracts (interphase extracts), in which protein phosphorylation mimics the one observed in G2 cells. We, therefore, added either *Xenopus* or human histidine recombinant FAM122A proteins to these extracts and measured the levels of PP2A-C and Wee1 over time. Surprisingly, we could not observe any variation of the amount of these two proteins in the extracts (Fig. 1A). We next proceeded by checking whether our recombinant FAM122A forms can inhibit PP2A-B55. To do this, we used interphase egg extracts in which ATP was eliminated and thus endogenous kinases are inactive (called hereafter as kinase-inactivated extracts). Because of the absence of endogenous active kinases, substrate dephosphorylation in these extracts directly reflects phosphatase activity(ies). Kinase-inactivated extracts were supplemented, or not, with the corresponding FAM122A recombinant protein (human/*Xenopus*) and subsequently with recombinant Arpp19 pre-phosphorylated at S113 by PKA known to be a bona fide PP2A-B55 substrate (Labbé et al, 2021; Lemonnier et al, 2019). Dephosphorylation of S113 of Arpp19 was then followed by western blot over time. As expected by its capacity to inhibit PP2A-B55, both human and *Xenopus* FAM122A drastically delayed Arpp19 dephosphorylation (Fig. 1B). Similar results were observed using phosphorylated-T481 PRC1, another well-established substrate of PP2A-B55 phosphatase (Cundell et al, 2013). We next checked the association of His-FAM122A to PP2A-B55 by using histidine-pull down. B55, A and C subunits of PP2A were present

in the histidine but not in the control pull-down of *Xenopus* and human FAM122A (Fig. 1C). Thus, FAM122A can directly bind and inhibit PP2A-B55 without any modification of the levels of the catalytic subunit of this phosphatase.

## FAM122A triggers mitotic entry by inhibiting PP2A-B55 and permitting cyclin A/Cdk-dependent phosphorylation of mitotic substrates

We next checked the impact of FAM122A on mitotic progression by measuring the phosphorylation of mitotic substrates upon the addition of this protein into interphase extracts. FAM122A induced a first phosphorylation of Gwl, the Anaphase Promoting Complex (APC) subunit Cdc27, and of Cdc25 concomitantly with the activation of cyclin B/Cdk1 as reflected by the loss of the phosphate on tyrosine 15 inhibitory site of Cdk1. This was followed by the subsequent degradation of cyclin A and B and the dephosphorylation of the above indicated substrates revealing that FAM122A promotes mitotic entry but is unable to maintain the mitotic state (Fig. 2A). We then sought to assess whether, as for Arpp19, FAM122A requires phosphorylation by Gwl to stably bind and inhibit PP2A-B55. Against this hypothesis, our data indicate that Gwl is unable to phosphorylate FAM122A "in vitro" (Fig. 2B) and the depletion of this kinase in interphase extracts did not prevent mitotic entry induced by FAM122A (Fig. 2C). Moreover, a FAM122A form in which all serine and threonine residues were mutated into alanine kept its capacity to induce mitosis (S/T-A FAM122A, Fig. 2D) and displayed a similar PP2A-B55 binding capacity than the wildtype FAM122A (Fig. 2E). These data suggest that FAM122A phosphorylation is not required for its binding to and the inhibition of PP2A-B55 at least at the used ectopic doses although we cannot exclude other PP2A-B55-independent role of this phosphorylation. Thus, FAM122A would not require either phosphorylation or Gwl activity to trigger mitotic entry.

We then investigated whether mitotic substrate phosphorylation induced by FAM122A requires cyclin B/Cdk1 activation by examining the effect of depleting Cdc25, the phosphatase dephosphorylating tyrosine 15 of Cdk1 and triggering cyclin B/Cdk1 activity. As expected, Cdc25-devoid extracts displayed inactive cyclin B/Cdk1, as stated by the presence of the phospho-tyrosine 15 on Cdk1 (Fig. 2F). Moreover, cyclin A became fully proteolyzed whereas cyclin B remained mostly stable although we detected a loss of a small fraction of this protein likely due to the phosphorylation, under these conditions, of Cdc27 and the activation of the APC. Besides Cdc27, FAM122A, via PP2A-B55 inhibition, was also able to promote the phosphorylation and activation of Gwl and, most importantly, the phosphorylation and inactivation of the Cdk1 inhibitory kinase Myt1. Since due to Cdc25 depletion cyclin B/Cdk1 was inactive in these extracts, we

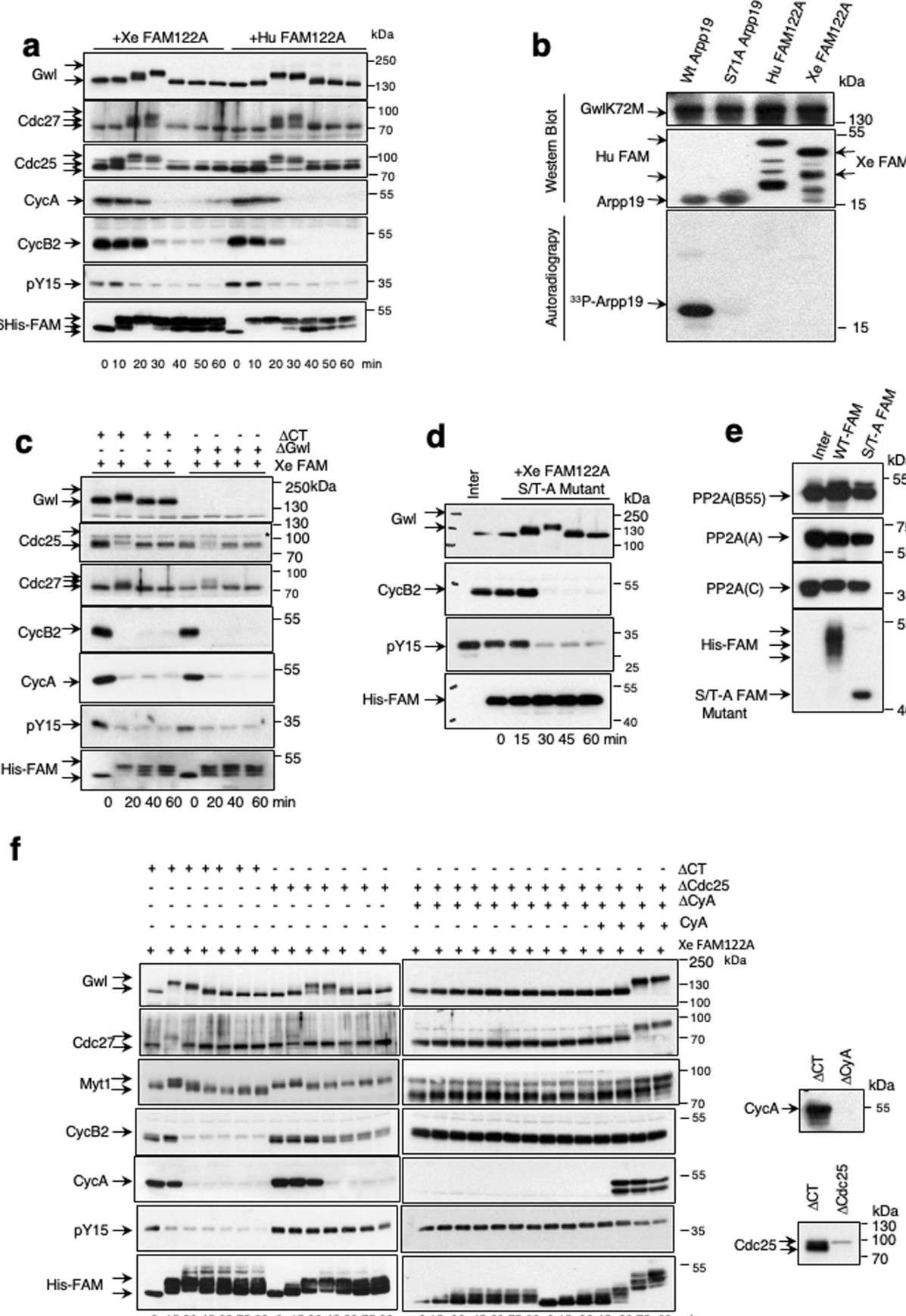

◄

**Figure 2. FAM122A triggers mitotic entry by inhibiting PP2A-B55 and permitting cyclin A/Cdk-dependent phosphorylation of mitotic substrates.**

(A) Extracts were treated as for Fig. 1A and the levels of the indicated proteins as well as the phosphorylation of Cdk1 on tyrosine 15 were checked by western blot. (B) His-wildtype Arpp19, the Gwl phosphorylation site mutant (Serine 71 -to-Alanine) of this protein, His-human FAM122A and His-*Xenopus* FAM122A were phosphorylated "in vitro" by GST-human GwlK72M hyperactive mutant in a final phosphorylation reaction mixture of 10 μl. 5 μl were then used for western blot using anti-histidine antibodies to detect Arpp19 and human and *Xenopus* FAM122A levels or anti-Gwl antibodies to detect Gwl amount. The rest was used to detect γ$^{33}$P by autoradiography. (C) Interphase extracts were depleted with a control or anti-Gwl antibodies and supplemented with Xe FAM122A. Samples were then analysed over time for the levels and phosphorylation of the indicated proteins. (D) Interphase extracts were supplemented with the *Xenopus* FAM122A protein in which all serine/threonine residues were mutated into alanine. Samples were recovered at the indicated time points and used for western blot. (E) Interphase extracts supplemented with a wildtype or a *Xenopus* FAM122A protein in which all serine/threonine residues were mutated into alanine and used for histidine pulldown and western blot to measure their association with B55, A and C subunits of PP2A. (F) Interphase extracts were depleted using control or anti-Cdc25 antibodies, or submitted to two consecutive depletions using firstly anti-Cdc25 and secondly anti-cyclin A antibodies. Depleted supernatants were then supplemented with Xe FAM122A protein. When indicated, purified cyclin A was added to the mix to a final concentration of 100 nM. Samples were recovered over time and used for western blot. Depletion of cyclin A and Cdc25 in the extracts was confirmed by western blot. Source data are available online for this figure.

attributed these phosphorylations to cyclin A/Cdk1. In this line, we previously showed that this complex escapes to Wee1-Myt1 inhibitory phosphorylation and is fully active in interphase extracts (Vigneron et al, 2018). To confirm this hypothesis, we co-depleted Cdc25 and cyclin A in FAM122A-supplemented interphase extracts. Remarkably, the loss of these two proteins prevented substrate phosphorylation resulting from FAM122A-dependent inhibition of PP2A-B55 (Fig. 2F). However, this phosphorylation was restored again if 45 min upon Cdc25/cyclin A co-depletion and FAM122A addition, cyclin A was supplemented again to these extracts. These data demonstrate that the decrease of the PP2A-B55 activity in interphase extracts induced by FAM122A is sufficient to permit cyclin A-Cdk1-dependent phosphorylation that will then trigger cyclin B/Cdk1 reactivation. Interestingly, this suggests that an increase of the endogenous levels of either FAM122A or cyclin A would be sufficient to trigger mitotic entry and would thus escape to the negative regulation of active PP2A-B55 during interphase.

## Molecular determinants of FAM122A controlling the association and the inhibition of PP2A-B55

AlphaFold artificial intelligence software (Alphabet's/Google's DeepMind) predicts the presence of two αhelices in the FAM122A protein, α-helix1: residues 84–93 (human)/71–82 (*Xenopus*) and α-helix2: residues 96–119 (human)/85–110 (*Xenopus*) (Fig. 3A). In order to investigate if these structured regions could participate to the association of this protein to PP2A-B55, we constructed a FAM122A *Xenopus* mutant forms deleted of either α-helix1 (αH1) or α-helix2 (αH2) and we measured their capacity to induce mitotic entry and to bind to PP2A B55 and C subunits when added to interphase extracts. Neither of these two mutants were able to induce mitosis (Fig. 3B) or to bind to PP2A-B55 (Fig. 3C) indicating that these two regions are essential for FAM122A inhibitory activity. We next constructed a N-terminal Δ(1–72) and a C-terminal (Δ110–270) mutant forms in order to determine whether any of these regions could additionally participate to the functionality of this protein. Unlike the two α-helix regions, these two mutants displayed similar properties than the wildtype form (Fig. 3D,E). Finally, we proceed with the construction of FAM122A single mutant forms for all conserved residues present in αH1 and αH2 regions. Data on mitotic entry and PP2A-B55 binding capacities of these mutants are shown in Appendix Fig. S1a,b and summarized in Fig. 3F. Residues of αH1, R73, L74, I77 and E80 were essential for FAM122A inhibitory activity. Indeed,

αH1 sequence SRLHQIKQEE, is close to the previous reported Short Linear Motif (SLiM) designed for PP2A-B55, with consensus sequence p[ST]-P-x(4,10)-[RK]-V-xx-[VI]-R(Fowle et al, 2021) and is highly conserved between species (Appendix Fig. S1c). This motif is present in PP2A-B55 substrates and is required for their binding to the phosphatase via their interaction with the B55 subunit. As so, these residues are involved in FAM122A association to B55. For αH2 region, we identified positions E90, H93, E94 and R95 as being essential for FAM122A activity. Thus, we pinpointed these two structured regions of FAM122A essential for its function and we identified the key residues directly mediating PP2A-B55 phosphatase interaction.

## Gwl activation at mitotic entry promotes the dissociation of FAM122A from PP2A-B55

Data above demonstrate that the addition of FAM122A to interphase extracts promotes mitotic entry but is unable to maintain mitotic substrate phosphorylation. Since the levels of FAM122A remain constant throughout the experiment and that FAM122A activity appears not to be impacted by phosphorylation, we sought to understand why FAM122A is not able to maintain the mitotic state. To this, we first checked the capacity of *Xenopus* and human FAM122A proteins to bind PP2A-B55 when they were supplemented to metaphase II arrested extracts (CSF extracts), which mimics the mitotic state. Interestingly, B55 and C subunit binding to FAM122A is clearly observed during interphase, whereas no association for B55 and a drastic decrease for C catalytic subunit were observed in CSF extracts (Fig. 4A).

To corroborate these observations, we tested whether FAM122A would be dissociated from B55 subunit in extracts blocked in mitosis. In order to obtain mitotic blocked extracts, we prevented cyclin B degradation in interphase extracts by the depletion of the APC subunit Cdc27 and then we added FAM122A. FAM122A addition promoted mitotic entry but, because of the incapacity to proteolyze cyclin B, extracts remained blocked in this phase of the cell cycle. Under these conditions, we measured the temporal pattern of protein phosphorylation and association of FAM122A to PP2A-B55. As expected, addition of FAM122A to control interphase extracts promoted mitotic entry and exit, while mitosis was maintained throughout the experiment when Cdc27 was depleted (Fig. 4B). In control extracts, FAM122A rapidly associated to B55, A and C subunits from its addition, then dissociated concomitantly with mitotic entry and re-associated again upon exit

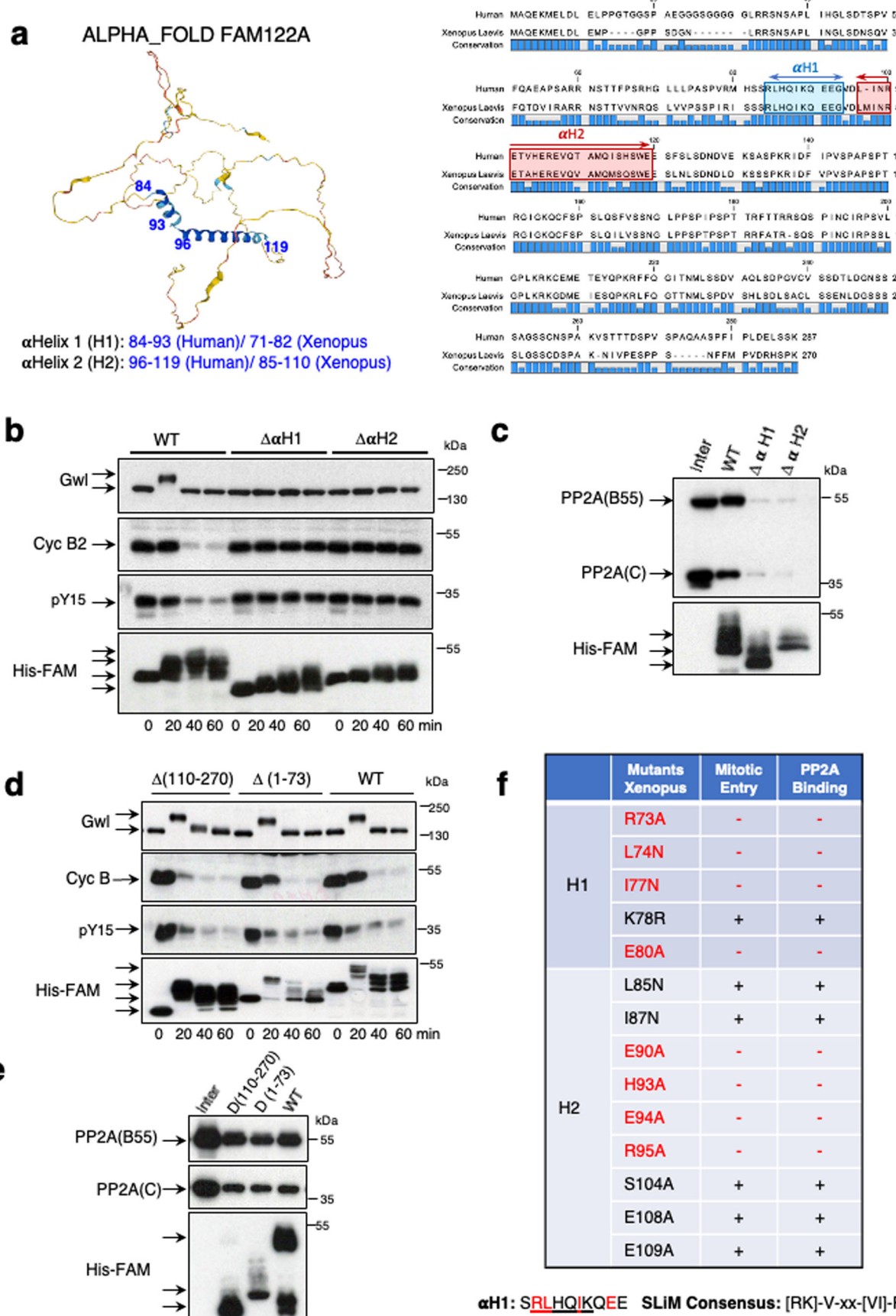

**Figure 3.  Molecular determinants controlling FAM122A inhibitory activity.**

(A) AlphaFold Monomer v2.0 predicts two structured αhelices (αHelix 1 and αHelix 2) in the human FAM122A protein and human. *Xenopus laevis* amino acid sequence alignment. αHelix 1 and αHelix 2 are indicated. (B) Interphase extracts were supplemented with the wildtype or the αH1 or αH2 mutants of Xe FAM122A and the phosphorylation and levels of the indicated proteins measured at the indicated times. (C) Interphase extracts supplemented with wildtype and αH1 or αH2 mutants were used for His-pulldown to measure the association of these proteins to B55 and C subunits of PP2A-B55. (D) The wildtype and the indicated mutant forms of Xe FAM122A were supplemented to interphase extracts and used for western blot. (E) His-pulldown was performed in extracts treated as in (D) to check the binding of the different forms of Xe FAM122A form to B55 and C. (F) Table indicating the capacity of the different single mutants of Xe FAM122A to induce mitotic entry and to bind to PP2A-B55. Mutants that have lost their inhibitory activity are depicted in red. The sequence of the αH1, the SLiM consensus sequence and the sequence of αH2 are shown. Residues essential to keep the inhibitory activity of Xe FAM122A are highlighted in red. Source data are available online for this figure.

of mitosis by cyclin B degradation (Fig. 4B). Conversely, in Cdc27-depleted interphase extracts, FAM122A also displayed a first association to these proteins but then, definitively lost this interaction from the establishment of the mitotic state. Taken together, these observations suggest that FAM122A-B55 binding is prevented during mitosis.

Besides, cyclin B/Cdk1, the Gwl kinase is also required to maintain the mitotic state. Although our data shown above demonstrate that Gwl cannot phosphorylate FAM122A to promote mitotic entry, we tested whether the PP2A-B55 inhibitory activity of FAM122A could be indirectly modulated by this kinase once the mitotic state is achieved. To this, we repeated the previous experiment except that extracts were immunodepleted of both, Cdc27 and the Gwl kinase. As expected by the depletion of Cdc27, all extracts entered into mitosis and kept the mitotic state as shown by the stability of cyclin B, the continuous phosphorylation of Cdc25 and the loss of phospho-tyrosine 15 Cdk1 signal throughout the experiment (Fig. 4C). Interestingly, unlike Cdc27/control-co-depleted extracts (Fig. 4C, lane 1, His Pull Down) and despite the presence of a fully active cyclin B/Cdk1 complex, FAM122A was able to bind to B55 and to increase its association to PP2A C subunit in Cdc27/Gwl-co-depleted extracts (Fig. 4C, lane 2, His Pull Down). However, FAM122A dissociated again from B55 when a Gwl hyperactive form was further supplemented (Fig. 4C, lane 4, His Pull Down). Thus, FAM122A dissociation from PP2A during mitosis is modulated by the Gwl kinase.

## Arpp19 competes and dissociates FAM122A from PP2A-B55 during mitosis

We next sought to decipher how Gwl could regulate FAM122A association to PP2A-B55. Gwl phosphorylates Arpp19 during mitosis, turning it into a high affinity interactor of PP2A-B55 (Mochida et al, 2010; Gharbi-Ayachi et al, 2010; Williams et al, 2014). We thus, asked whether during mitosis, phosphorylated Arpp19 competes with FAM122A for PP2A-B55 binding. To test this hypothesis, we supplemented interphase extracts with FAM122A and 40 min later, once the extract returned to an interphase state and FAM122A binds again PP2A-B55 (Figs. 2A and 4B), it was either (1) used to perform a histidine-FAM122A pulldown and subsequently supplemented with thio-S71-phosphorylated Arpp19 or the other way around, (2) first supplemented with a thio-S71-phosphorylated Arpp19 and subsequently submitted to histidine-FAM122A pulldown (Fig. 5A, scheme). Finally, FAM122A association to B55 was measured. Strikingly, the presence of a thio-S71-phosphorylated form of Arpp19 (Thio-S71-Arpp19) that cannot be dephosphorylated,

induced the dissociation of FAM122A from PP2A-B55 in both cases confirming our hypothesis that Arpp19 displaces FAM122A from PP2A-B55 by competing for this phosphatase.

We also performed the reverse experiment and tested whether GST-FAM122A could dissociate a p-S71-Arpp19 complexed to PP2A-B55 phosphatase. Thus, CSF extracts were supplemented with His-Arpp19 protein. Because of the activated Gwl kinase present in these extracts, exogenous Arpp19 is immediately phosphorylated and tightly binds PP2A-B55. p-S71-Arpp19-PP2A-B55 complex was then recovered by histidine-pulldown and supplemented with high doses of GST-FAM122A. Then the level of B55 bound to pArpp19 was measured by western blot. Interestingly, the levels of B55 bound to pArpp19 did not change upon FAM122A addition indicating that FAM122A was unable to dissociate this protein from PP2A-B55 complex (Fig. 5B). These data indicate that p-S71-Arpp19 has a much higher affinity for PP2A-B55 than FAM122A and induces its dissociation. Supporting this hypothesis, our data above (Figs. 4 and 5B) demonstrate that endogenous Arpp19 is able to dissociate ectopic His-FAM122A protein from PP2A-B55 despite we estimated the molarity of the former in these experiments being 215 times lower than the latter (0.1 μM endogenous Arpp19 versus 21.5 μM ectopic His-FAM122A, Appendix Fig. S2a,b).

To deeper address the different binding affinities of pArpp19 and FAM122A for PP2A-B55, we performed kinetic studies directed to identify the inhibitor constants ($K_i$) of these two proteins by measuring dephosphorylation of p-S113-Arpp19 in kinase inactivated extracts supplemented with increasing doses of Thio-S71-Arpp19 or FAM122A. Inhibition of PP2A-B55 by FAM122A and Thio-S71-Arpp19 was characterized by varying both the concentration of the substrate p-S113-Arpp19 (from 120 to 2000 nM) and that of the inhibitors FAM122A (from 0 to 250 nM) or Thio-S71-Arpp19 (from 0 to 3 nM). Of the four inhibition modes tested, the competitive mode was the one fitting the experimental data with the lowest reduced Chi$^2$ value, which reflects the fitting quality (Appendix Figs. S3, S4). The values of the $K_i$ were 28.1 ± 4.6 nM for FAM122A and 0.22 ± 0.04 nM for Thio-S71-Arpp19 with a consistent Km for p-S113-Arpp19 of around 145 ± 30 nM in both series of experiments (Fig. 5C and Appendix Figs. S3, S4). The $K_i$ for FAM122A was 127-fold higher than the one observed for Thio-S71-Arpp19, an affinity value very close to the one that we obtained by measuring p-S71-Arpp19-dependent dissociation of FAM122A in pulldown assays (215-fold higher). Thus, these data confirm the higher affinity of p-S71-Arpp19 for PP2A-B55 and its capacity to dissociate FAM122A of this phosphatase.

We next checked whether the temporal patterns of (1) association and dissociation of FAM122A/PP2A-B55, (2) activation

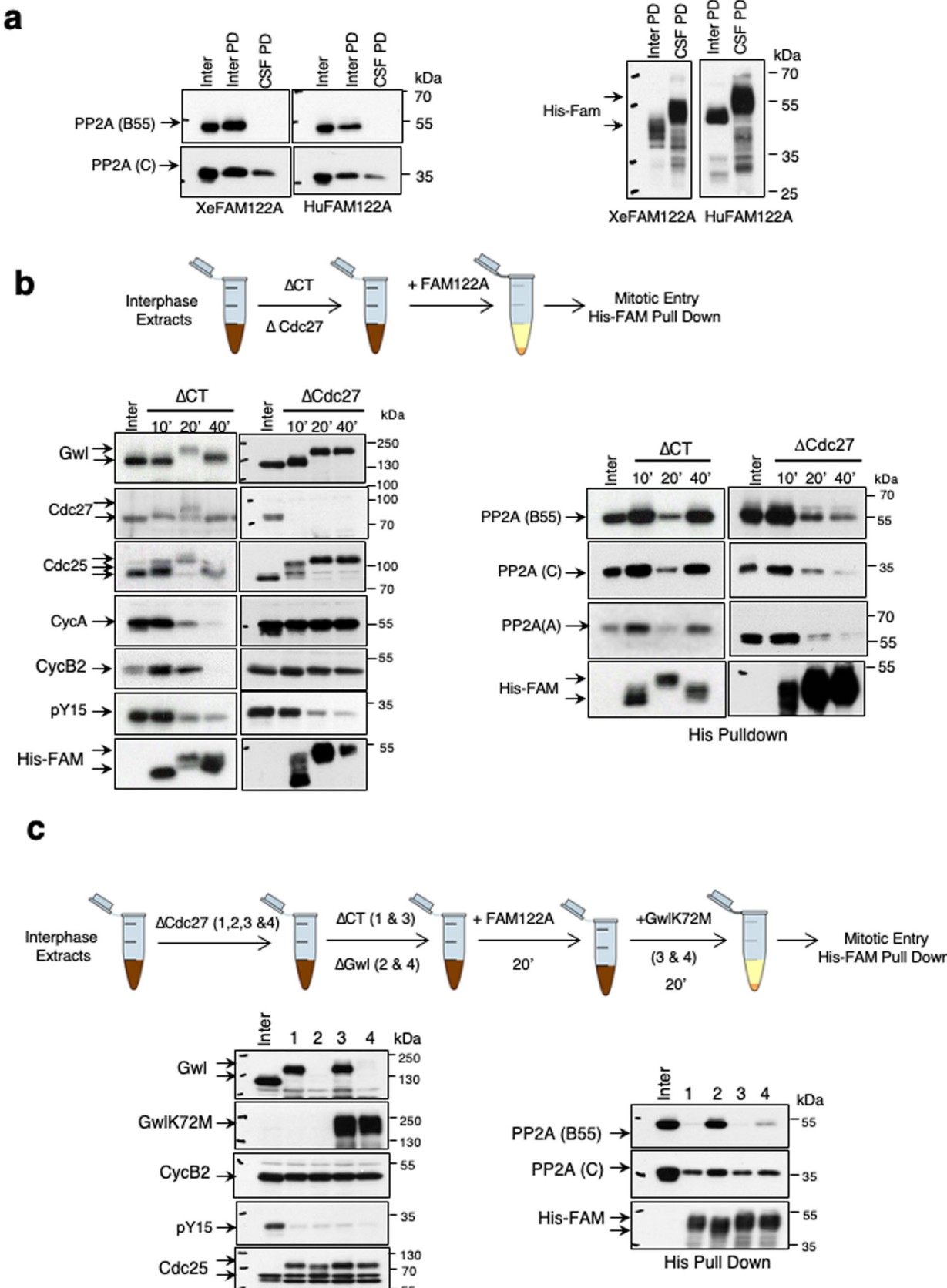

◀ Figure 4. Gwl activation at mitotic entry promotes the dissociation of FAM122A from PP2A-B55.

(A) Interphase and CSF extracts were supplemented with Xe FAM122A and Hu FAM122A and used for His-pulldown to measure the association of this protein with B55 and C subunits of PP2A-B55. The levels of His-Xe/HuFAM122A recovered in the pull down are also shown. Inter PD: Interphase Pull Down. CSF PD: CSF pull down. (B) Interphase extracts were depleted with control or with anti-Cdc27 antibodies and subsequently supplemented with Xe FAM122A and used for western blot or for His-pulldown to assess the binding of Xe FAM122A to PP2A-B55. (C) Interphase extracts were depleted of Cdc27, submitted to a second immunodepletion using control (lanes 1 & 3) or anti-Gwl antibodies (lanes 2 & 4) and supplemented with Xe FAM122A. 20 min later Gwl activity was restored (lanes 3 & 4) or not (lanes 1 & 2) by adding a hyperactive form of Gwl (GwlK72M). Samples were used for western blot and His-pulldown. Source data are available online for this figure.

of Gwl and (3) Gwl-dependent phosphorylation of Arpp19 in S71 conferring its PP2A-B55 inhibitory activity, were in line with the differential $K_i$ constants obtained for FAM122A and Thio-S71-Arpp19. Accordingly, the addition of FAM122A to interphase extracts promoted the immediate presence of B55 in FAM122A pulldowns (10 min) and its subsequent dissociation concomitantly with Gwl activation (10 and 20 min) and Arpp19 phosphorylation (20 and 30 min) (Fig. 5D). However, this protein was re-associated again once cyclin B was degraded, Gwl was inactivated and Arpp19 was dephosphorylated at mitotic exit (40 min). These data support that Arpp19 phosphorylation during mitosis promotes the dissociation of FAM122A from PP2A-B55 and takes over the inhibition of this phosphatase. However, to fully confirm the physiological role of FAM122A in mitotic entry, it is essential to show that ectopic FAM122A doses supplemented to these extracts are close to the endogenous concentration of this inhibitor. We thus tried to measure the endogenous concentration of FAM122A in *Xenopus* extracts. However, unfortunately, although our antibodies recognize the ectopic protein, they were unable to detect FAM122A in the extracts. Since this protein is nuclear (Li et al, 2020), we reasoned that it was lost during the purification of oocyte cytoplasmic fraction. As an alternative, we decided to compare the minimal dose of FAM122A used in the extracts to the concentration of endogenous FAM122A in human cells. By comparing western blot signals of recombinant His-FAM122A and RPE1 cell lysates we estimated endogenous FAM122A concentration to be $59.6 \pm 6.37$ nM (Appendix Fig. S5). We next performed dose-response assays to identify the minimal dose of FAM122A inducing mitotic entry when supplemented to interphase extracts using the wildtype and the phosphorylation S/T-A mutant FAM122A forms. We could establish that a dose between 3 and 15 nM of ectopic wildtype FAM122A was sufficient to promote mitotic entry (Fig. 5E). Conversely, the S/T-A FAM122A mutant minimal dose inducing mitotic entry was fixed between 30 and 75 nM suggesting that the phosphorylation of this protein exerts a positive role, but is not essential for PP2A-B55 inhibition. Moreover, although higher than the one obtained for the wildtype protein, S/T-A FAM122A mutant minimal concentration is very close to the one estimated for endogenous FAM122A in human cells (30 to 75 vs 59.6 nM). All together these data are in line with the first role of FAM122A in PP2A-B55 inhibition at mitotic entry that would be subsequently taken over by Arpp19 once phosphorylated by active Gwl.

## The different binding of Arpp19 and FAM122A to PP2A-B55 could explain the ability of the first inhibitor to dissociate the second from phosphatase

In order to understand how Arpp19 can dissociate FAM122A from PP2A-B55, we modelled the quaternary complex of PP2A with

Arpp19 from three distant species (human, *Xenopus* and *C. elegans*) using Alphafold_multimer (preprint: Evans et al, 2021) with default settings and no relaxation. The resulting models looks globally similar. The molecular organization of the PP2A subunits reproduces well the known crystal structure (PDB3DW8) with little unfolded regions. Arpp19 (for the human and *Xenopus* complexes) and the unique orthologue for Arpp19/ENSA in *C. elegans* called ENSA are mostly unfolded (Fig. 6A, left panel, orange) and share three common helical stretches, labelled hereafter as α1, α2 and α3. A first helix, α0, in human Arpp19 and *Xenopus* Arpp19, that is not predicted in worm ENSA, seems to form no interaction with PP2A and will not be further considered. On the contrary, the three conserved helices (α1: residues 24–35, α2 :45–54 and α3 :60–73 in human Arpp19) are predicted to interact with the phosphatase. The first two contact the B55 subunit (Fig. 6A, left panel, blue) while the third one points toward the catalytic site of the C subunit (Fig. 6A, left panel, green). No contact is detected between the Arpp19 chains with the scaffolding subunit (Fig. 6A, left panel, violet). The interaction with B55 mainly relies on the helix α1 that displays tight contacts along the helix from Arpp19. The interactions conserved among the three complexes includes two salt bridges (E25 and R35 from Arpp19 with R330 and E338 from B55 in the human complex) and hydrophobic interactions involving L32 from Arpp19 and F280, F281 and C334 from B55 subunit in human. In addition, two N-capping and one C-capping (involving E28, E29 and K26 in human and Arpp19, respectively) are predicted. The second helix of Arpp19, α2, shows more loose interactions and it seems to mainly bridge the helix α1 with the third helix. Helix α3 partially overlaps with the QKYFDSGDY motif conserved in all Arpp19 and ENSA sequences and containing the serine phosphorylated by Gwl. This phosphorylation is essential for strong inhibition of PP2A. We modelled the phosphoserine using a dedicated server (https://isptm2.cbs.cnrs.fr described in the "Methods" section) and also added the two manganese atoms within the catalytic center of the C subunit of PP2A for further analysis. Although in the human and *Xenopus* complexes, the serine is not predicted to directly interact with the catalytic manganese (distance PO4-Mn: ~8 A), such interactions appeared plausible in the worm complex (distance PO4-Mn: ~5 A). This direct contact would explain the strong interaction of phosphorylated Arpp19 with PP2A-B55, although it is worth highlighting that the $Mn^{2+}$ ion present in the structure obtained with recombinant PP2A-B55 enzyme does not correspond to the $Zn^{2+}:Fe^{2+}$ divalent cation pair known to be present in the native PP2A-B55 enzyme (Brautigan and Shenolikar, 2018). However, the inhibitory activity has been shown to be not only characterized by the high affinity of Arpp19 towards PP2A-B55, but also by the slow dephosphorylation of its Gwl site (Labbé et al, 2021). The present modelling does not permit to fully understand

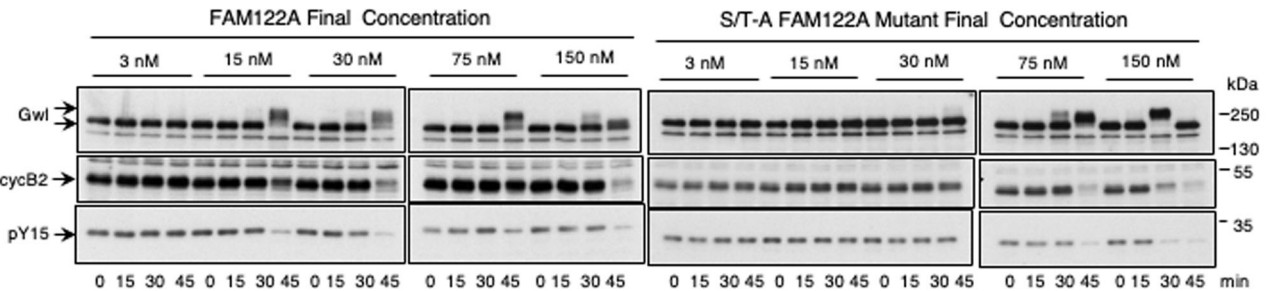

Table c:

| | $V_{max}$ (nM) | $K_m$ (nM) | $K_i$ (nM) |
|---|---|---|---|
| pS113-Arpp19 | 90±4 | 145±30 | |
| pS67-Arpp19 | | | 0.22±0.04 |
| FAM122A | | | 28.1±4.6 |

◄ **Figure 5. Arpp19 dissociates FAM122A from PP2A-B55 during mitosis.**

(A) Interphase extracts were supplemented with a final concentration of 7.15 µM of Xe FAM122A and 40 min later divided into two samples. One of these samples was used to perform a His-pulldown that was then supplemented with a final concentration of 2.7 µM of untagged Thio-Arpp19 protein and used to measure bound B55 protein. The second sample was firstly supplemented with untagged 69.3 pmol of Thio-Arpp19 and subsequently used for His-pulldown to measure His-Xe FAM122A bound B55 subunit of PP2A. A sample of the His-Xe FAM122A pull downs supplemented (PD+Arp) or not (PD-Arp) with untagged Thio-Arpp19 is shown. (B) CSF extracts were supplemented with a final concentration of 3,6 µM His-Arpp19 and, 40 min later, used for His-pulldown. Arpp19-pull downs were then supplemented with a final concentration of 10 µM GST-Xe FAM122A and the levels of B55 remaining were measured. A sample of the His-Arpp19 pull downs supplemented (PD + FAM) or not (PD-FAM) is shown. (C) Table indicating the kinetic parameters of the dephosphorylation of the PP2A-B55 substrate p-S113-Arpp19 and the inhibitory constant of FAM122A and Thio-S71-Arpp19 in kinase inactivated interphase extracts. Conditions used in these assays are detailed in Methods. (D) Interphase extracts were supplemented with His-Xe FAM122A and used for either western blot to measure the levels and the phosphorylation of the indicated proteins and or for His-pulldown determine Xe FAM122A binding to PP2A B55, A and C subunits at the indicated time points. (E) Interphase extracts were supplemented with His-FAM122A to the indicated final concentrations and mitotic entry was assessed by the phosphorylation of Gwl, dephosphorylation of tyrosine 15 of Cdk1 and the degradation of cyclin B2. Source data are available online for this figure.

the absence of catalytic activity on the phosphorylated serine. However, this model agrees with the previously reported mutational scanning of the QKYFDSGDY motif (Labbé et al, 2021) to suggest that a precise and peculiar orientation of the phosphorylated sidechain would be required to maintain a tight but unreactive interaction. Indeed, mutations to alanine of close phosphoserine neighbour residues in this sequence dramatically accelerate Arpp19 dephosphorylation by PP2A-B55 while lowering binding at the same time (Labbé et al, 2021; Williams et al, 2014). Additional interactions are predicted at this interface in the worm complex (and partially reproduced in the human and *Xenopus* assemblies) (see Fig. 6B). These interactions include two salt-bridges involving the Gwl site phosphoserine (S62) and the preceding aspartate residue (D61) from human Arpp19 and two arginines of human PP2A C subunit (R89 and R214) lying at the entrance of the catalytic site. Another salt bridge may occur between another aspartate from Arpp19 (D64 in human Arpp19) and a lysine (K88) in the human B55 subunit. In addition, in the worm complex, we observed the nearby and conserved phenylalanine F65 (F60 in human Arpp19) stacked in between the subunits B55 and C. This residue seems to make hydrophobic contact with a leucine L126 of B55 (L87 in human B55). Other contacts are not systematically predicted in the three complexes but they may enhance the overall interaction between Arpp19 and the PP2A phosphatase. Noteworthy, the three conserved residues surrounding the phosphorylated serine, F60, D61 and D64 in human Arpp19, seems important for precise positioning of the serine into the catalytic site of PP2A. This nicely explained the impact of their mutation to alanine, that produce variants that are rapidly dephosphorylated by PP2A-B55 and that lose their affinity for the phosphatase (Labbé et al, 2021).

Interestingly, no complexes could be predicted for the full-length FAM122A inhibitor with PP2A-B55 using Alphafold 2.2. On the contrary, short segments of FAM122A (from the three species used above) led to prediction of interactions between FAM122A and PP2A-B55. The interface partially overlaps with that observed for PP2A-B55-Arpp19 complexes supporting the fact that the two inhibitors compete for PP2A-B55 binding. Nevertheless, the precise interactions differ significantly. First, FAM122A does not seem to point into the catalytic site as Arpp19 seems to do, although there are also two predicted salt bridges between two glutamate residues from FAM122A (E100 and E104 in human FAM122A) with the same arginines contacting D61 and S62 of Arpp19 of the catalytic subunit of PP2A (R89 and R214 in human PP2A) (Fig. 6B). These two interactions are crucial since, as shown above (Fig. 3F), the

corresponding E90A and E94A *Xenopus* FAM122A mutants are unable to bind the phosphatase. Finally, as expected, the second helix of FAM122A containing the putative conserved SLiM motif, interacts with the B55 subunit and this interaction takes place in a region of B55 close to the one binding α2 helix of Arpp19.

In conclusion, our modelling data obtained for Arpp19 and FAM122A interaction with PP2A-B55 using Alphafold multimer suggest that the two inhibitors would produce analogous but distinct complexes with their common target. The size of these interfaces suggests that Arpp19 would be a stronger inhibitor than FAM122A and could thus promote the dissociation of the latter. This assumption is supported by newly predicted contacts with PP2A-B55 likely playing a major role in the strong inhibition of this enzyme by Arpp19 and that include the phosphorylation of the central serine that would tightly interact with the catalytic manganese ions.

## FAM122A is required "in vivo" for mitotic entry in the *C. elegans* germline

Our data above using ectopic FAM122A demonstrates that the addition of this protein to interphase *Xenopus* egg extracts promote mitotic entry. We thus sought to assess the impact of the depletion of the endogenous FAM122A from these extracts in mitotic entry. However, unfortunately, probably because we lost FAM122A during cytoplasmic extract purification, we were unable to immunoprecipitate endogenous FAM122A. We thus used the nematode *Caenorhabditis elegans* as a model to investigate the role of FAM122A in PP2A-B55 inhibition and mitotic entry. The gene *F46H5.2* in *C. elegans* has been previously suggested as the putative orthologue of FAM122A protein (Kim et al, 2018). To confirm that it does indeed correspond to *C elegans* FAM-122A (Ce FAM-122A), we produced and purified this protein and we checked whether it was able to inhibit PP2A-B55 and to promote mitotic entry in interphase extracts. As shown in Fig. 7A, dephosphorylation of S113 of Arpp19 and of T481 of PRC1 by PP2A-B55 was strongly delayed by the addition of this protein. Moreover, when supplemented to interphase extracts, Ce FAM-122A induced mitotic entry and exit (Fig. 7B). Thus, the product of the gene *F46H5.2* does correspond to the FAM122A orthologue.

We then investigated the capacity of endogenous Ce FAM-122A to inhibit PP2A-B55 "in vivo" in the worm. To this, we first took advantage of a multivulva phenotype induced by the mutant *let-60(n1046gf)*. This mutant, corresponding to a gain of function of

**a**

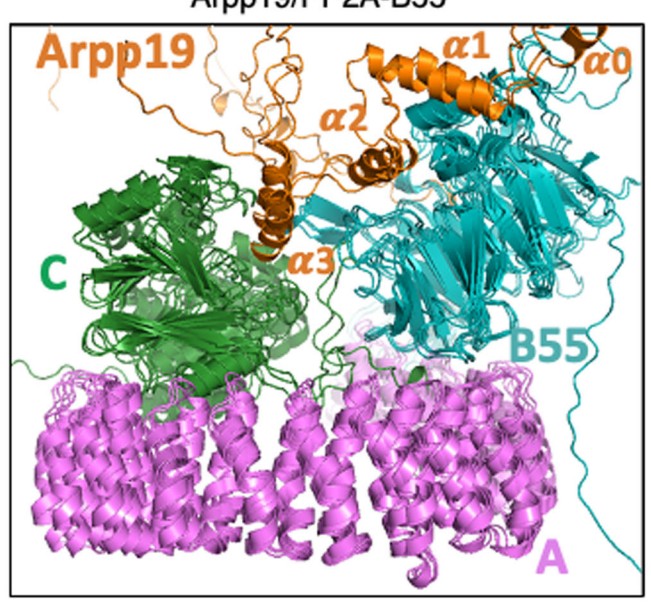
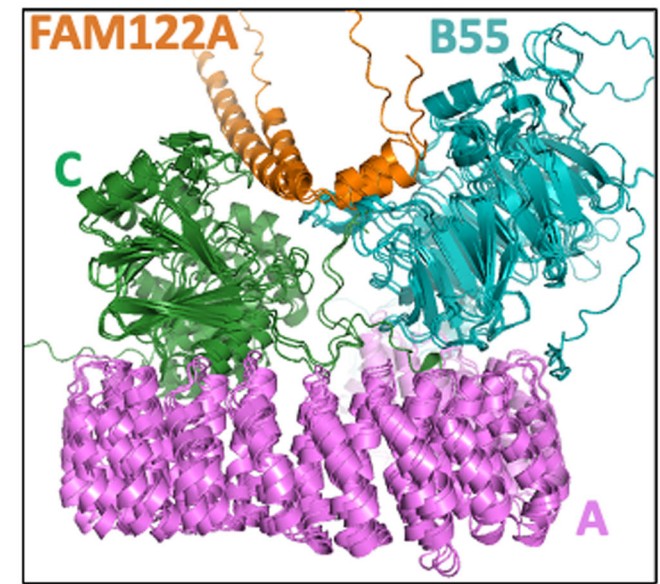

**b**

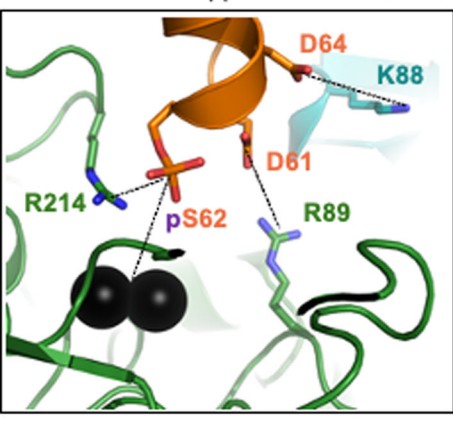
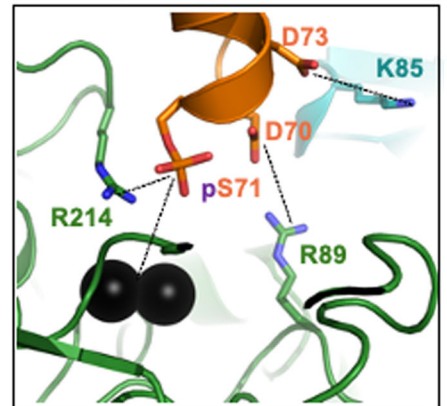
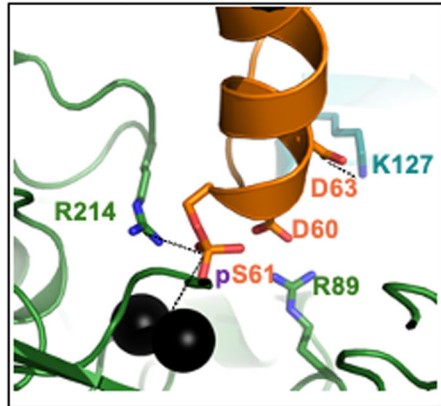

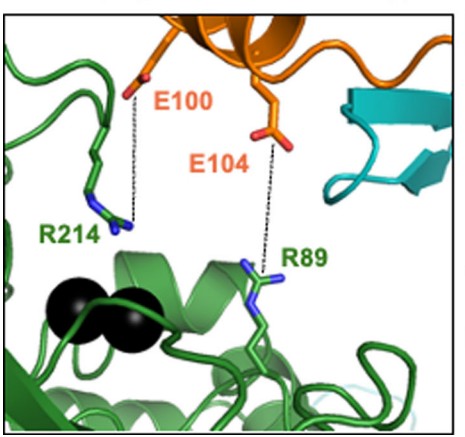
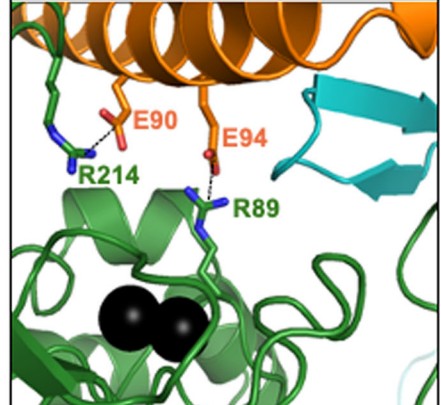
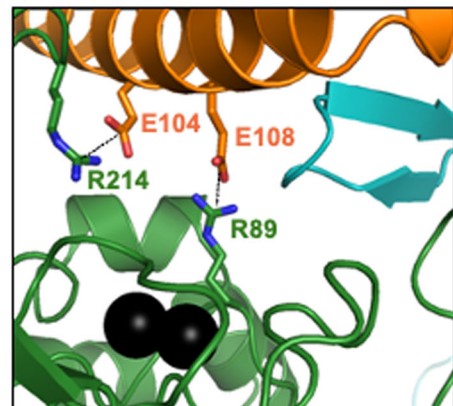

**Figure 6. Arpp19 and FAM122A differently bind PP2A-B55.**

(A) Superposition of the models of Arpp19/ENSA and FAM122A in complex with ternary PP2A-B55 as predicted by Alphafold_multimer v 2.2 for three species (human, *Xenopus* and *C. elegans*). Arrp19/ENSA and truncated FAM122A are in orange ribbon while PP2A-B55 is in blue (B55), dark green (catalytic C subunit) and pink (the scaffold A subunit). (B) Zoom onto the entrance of the catalytic site of PP2A in complex with human Arpp19 and FAM122A. Manganese ions are shown as black spheres and sidechains are drawn as sticks with carbon colours as the corresponding polypeptide chains. Figure was drawn using Pymol (http://www.pymol.org).

the human orthologue Ras, promotes the formation of several vulvas in the worm (Fig. 7C). It has been shown that the multivulva phenotype induced by *let-60(n1046gf)* (Sieburth et al, 1999) can be attenuated by negatively modulating PP2A-B55 activity as this protein promotes Ras signalling during vulva development. Accordingly, we observed a suppression of the multivulva phenotype after depletion of SUR-6 (SUpressor of Ras-6, the orthologue of human B55 subunit of PP2A) by RNAi. In order to assess whether endogenous Ce FAM-122A acts as an inhibitor of PP2A-B55, we examined the effect of the depletion of this protein by RNAi on the multivulva phenotype induced by the *let-60(n1046gf)* mutant. If this protein is a PP2A-B55 inhibitor "in vivo", its depletion should promote the reactivation of this phosphatase and thus significantly increase the presence of multivulvas in the worms. Confirming this hypothesis, the depletion of Ce FAM-122A by RNAi dramatically increased this phenotype (Fig. 7C). Thus, Ce FAM-122A does act as an inhibitor of PP2A-B55 "in vivo".

We further explored the role of this protein in *C. elegans* in mitosis "in vivo", in *F46H5.2^FAM122A^(RNAi)* treated worms. Interestingly these worms exhibit a decreased fecundity (Fig. 7D) with a significant increased percentage of embryonic lethality at 25 °C (median: 8.4% in RNAi-treated vs 0.30% in controls; $p < 0.0001$) and a double number of unfertilized eggs than controls (median 26.18% in RNAi-treated vs 12.10% in controls; $p < 0.014$) (Fig. 7E). Even wildtype *C. elegans* can produce unfertilized eggs when they age however, *F46H5.2^FAM122A^(RNAi)* worms display this phenotype on their first day of adulthood. These results suggest a fertility defect and led us to explore the function of the reproductive system. We thus investigated cell division within the *C. elegans* germline using nematode strains expressing histone and gamma-tubulin tagged with GFP under a germline and embryonic promoter. We focused on mitotic division of the germline stem cells of the distal gonad. Using chromatin and centrosome as proxy of mitotic progression (Gerhold et al, 2015), we counted the number of stem cells in interphase and in mitosis (Appendix Fig. S6). We observed a significant reduction of interphase cells and an increase in mitotic cells in RNAi treated worms (Fig. 8A). Interestingly, the number of cells in prophase (assessed by the presence of two separating centrosomes and a circular intact nuclei), was also dramatically increased, although some cells could progress into mitosis, probably due to a partial depletion of FAM-122A. However, these cells displayed a perturbed mitotic progression since we observed a drop of the number of anaphases (Fig. 8B). These data suggest that depletion of *F46H5.2* gene product decreases the capacity of germline stem cells to enter into mitosis and perturbs their progression through this phase of the cell cycle. To further investigate this hypothesis, we used a cell line with GFP-tagged β- or α-tubulin and with RFP-tagged histone in which we followed by time lapse microscopy the duration of mitosis. We considered mitosis duration as the time from first centrosome separation

movement starts to anaphase onset. As depicted in Fig. 8C, stem cells from control worms displayed a rapid mitosis, with a timing around 21 min. Conversely, most Ce FAM-122A depleted stem cells, stayed with separated centrosome throughout the time of the experiment (40–50 min) indicating that these cells were arrested or highly delayed in prophase. We observed fewer cells entering mitosis in the FAM-122A depleted worms (Fig. 8C, upper graph). In addition, although they did not display differences in the timing from nuclear envelope breakdown to metaphase (congression), they showed a significant increase of the time required for anaphase onset (Fig. 8C, lower graph). This delay is in accord with the drop of the number of anaphase in RNAi-treated germline stem cells and suggests that FAM122A-dependent inhibition of PP2A-B55 could not be only required for mitotic entry, but could also be essential again during mitotic exit to promote anaphase onset when Gwl and cyclin B/Cdk1 become inactivated by cyclin B proteolysis.

## Endogenous FAM122A proteins accumulates during G2 in human cells

How cyclin B/Cdk1 is triggered at G2-M is a main question yet to be answered. To activate cyclin B/Cdk1, Wee1/Myt1 and Cdc25 have to be phosphorylated. To that, an increase of the activity of cyclin A/Cdk, as well as a drop of PP2A-B55 are essential. The fact that FAM122A does not require phosphorylation to inhibit PP2A-B55 at the endogenous concentration and that cyclin A/Cdk is a poor substrate of Myt1 kinase during G2 (Coulonval et al, 2003; Booher et al, 1997) make of these two factors good candidates for triggering mitosis. However, to promote G2-M transition, it is essential that these two proteins accumulate during G2 to reach sufficient levels to revert cyclin B/Cdk1 inhibition. Accumulation of cyclin A has been largely demonstrated in the bibliography. Much less is known about the levels of FAM122A during G2. We thus measured the levels of FAM122A throughout the cell cycle in RPE1 cells. Interestingly, as depicted in Fig. 8D, FAM122A protein accumulates during G2 reaching maximal level at G2-M transition concomitantly with cyclin A accumulation. Moreover, the amount of these two proteins rapidly decrease during mitotic progression supporting the hypothesis of a putative role of these two proteins in triggering cyclin B/Cdk1 activation and mitotic entry.

## Discussion

Protein phosphorylation plays a major role in the control of cell cycle progression. This phosphorylation results from a fine-tuned balance between the activities of cyclin/Cdk complexes and phosphatases. The modulation of the phosphatase PP2A-B55 is key for S and M phases. During DNA replication, PP2A-B55 has to be inhibited to maintain the phosphorylation of the replication factor Treslin responsible of origin firing, while at G2-M, the

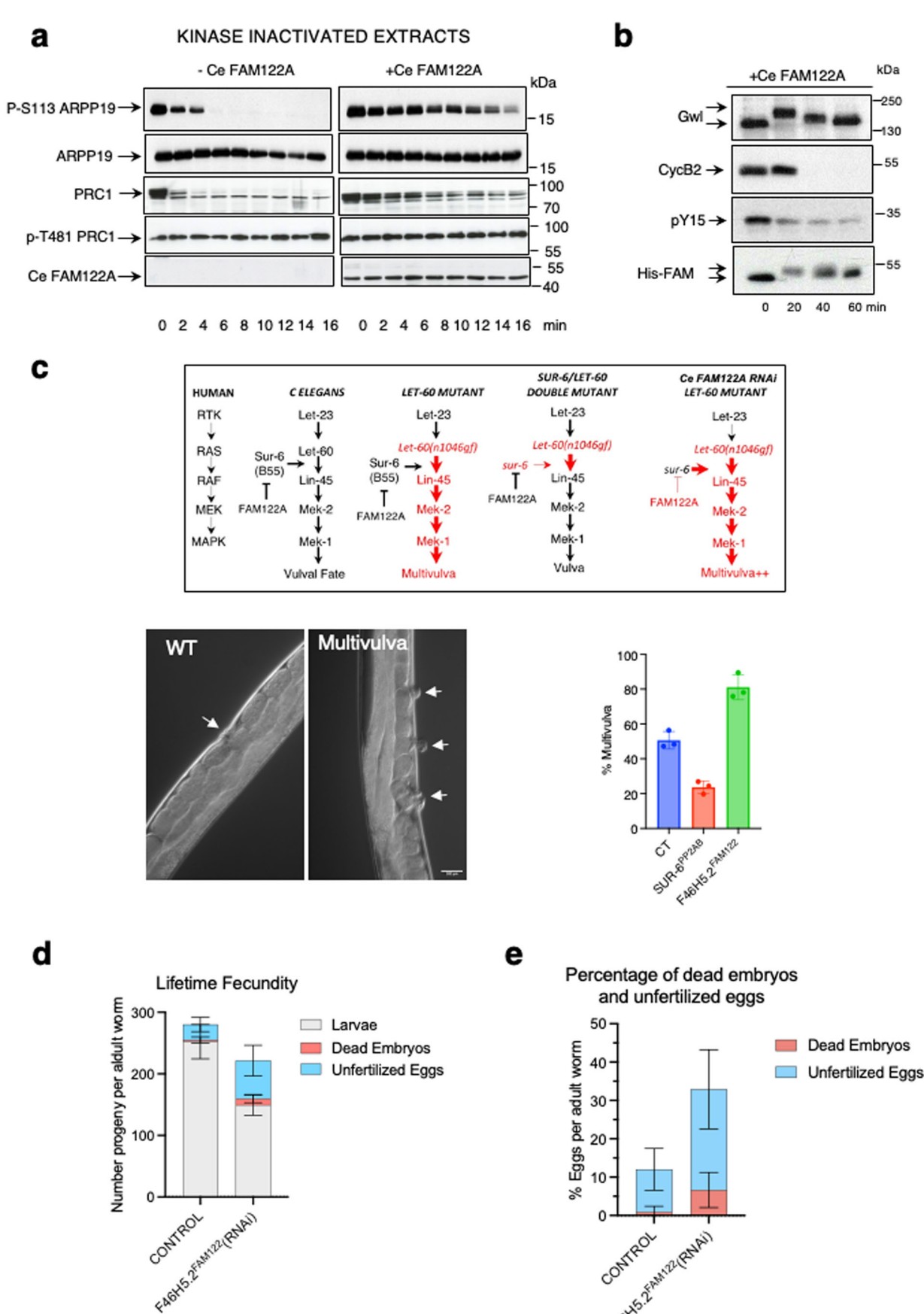

◄ **Figure 7.  FAM122A is required to correctly enter into mitosis in *C elegans*.**

(A) Arpp19 or PRC1 "*in vitro*" phosphorylated by PKA or Cdk1 respectively were supplemented together with Ce FAM122A to interphase extracts and the levels and dephosphorylation rate of S113 of Arpp19 or T481 of PRC1, as well as the amount of Ce FAM122A, were analysed by western blot. (B) Ce FAM122A was added to interphase extracts and the levels and phosphorylation of the indicated proteins measured by western blot. (C) Scheme representing the genetic dependences of the gain of function *let-60(gf)* mutant and *sur-6* (B55 orthologue) mutant on multivulva phenotype in *C. elegans*. *sur-6* mutant (diminishes PP2A-B55 activity) on a *let-60(gf)* mutant background decreases the multivulva phenotype. Conversely, the depletion of *F46H5.2* (FAM122A orthologue) in a *let-60(gf)* mutant background significantly increases this phenotype suggesting that, as expected, *F46H5.2* protein would act as an inhibitor of PP2A-B55 in *C. elegans*. Big red arrows: increased multivulva phenotype. Shown are two representative images of a wildtype worm displaying one vulva or a mutant worm with a multivulva phenotype. The percentage of worms displaying multivulva phenotype in each mutant is represented as a mean $+/-$ standard deviation. Significant differences calculated by non-parametric two tailed Mann-Whitney test are shown. (D) The number of larvae, dead embryos and unfertilized eggs were counted for control and *F46H5.2(RNAi)* worms and represented as mean values $+/-$ standard deviation. (E) The percentage of dead embryos and unfertilized eggs were counted in worms at first day of adulthood and represented as for (D). Source data are available online for this figure.

negative modulation of this phosphatase is essential for cyclin B/Cdk1 activation.

To enter into mitosis, a partial pool of cyclin B/Cdk1 has to be turned on to induce the amplification loop that triggers cyclin B/Cdk1 full activation via Wee1/Myt1/Cdc25 phosphorylation. However, Wee1/Myt1/Cdc25 phosphorylation can be also achieved if PP2A-B55 is at least partially inhibited. This inhibition depends on the phosphorylation of Arpp19/ENSA and thus, requires the activity of its upstream kinase Gwl. However, at M entry, both Gwl and cyclin B/Cdk1 are inactive and their activation depends on each other. Thus, none of them can be the initial event triggering mitosis.

In this study we investigated the role of FAM122A, a new identified regulator of PP2A-B55, in triggering mitotic entry. Using *Xenopus* egg extracts we demonstrate that the negative regulation of PP2A-B55 by FAM122A does not induce Wee1 or PP2A catalytic subunit proteolysis as previously proposed (Fan et al, 2016; Li et al, 2020). Conversely, it promotes mitotic entry by directly binding and inhibiting the activity of this phosphatase. PP2A-B55 inhibition by FAM122A precedes Gwl and cyclin B/Cdk1 activation since its phosphorylation is not required to negatively modulate the phosphatase, although we cannot exclude an additional positive modulation of its activity by this posttranslational modification. However, FAM122A-dependent induction of mitosis depends on cyclin A/Cdk, indicating that FAM122A promotes M-Phase entry by promoting cyclin A/Cdk-dependent phosphorylation. We, thus, hypothesize that the initial event triggering mitotic entry corresponds to the accumulation of FAM122A and cyclin A/Cdk1 that would result in a first burst of FAM122A-dependent inhibition of PP2A-B55 and the subsequent cyclin A/Cdk1-dependent phosphorylation of Cdc25 and Myt1. This will in turn result in the full activation of cyclin B/Cdk1 and mitotic entry (Fig. 9). Accordingly, our data show that, as for cyclin A/Cdk, FAM122A protein accumulates during G2, reaching a maximum level at G2-M transition. We additionally show that FAM122A plays a key role in PP2A-B55 inhibition and mitotic entry "in vivo" in the *C. elegans* model in which, FAM122A knockdown modulates phosphatase activity during vulval development and prevents mitotic entry in germ line stem cells.

Remarkably, a very low dose of FAM122A (15 nM) is sufficient to promote mitotic entry when added to interphase extracts. This dose would correspond to about 15% of the total concentration of PP2A-B55 present in these extracts (100–300 nM (Mochida et al, 2009)) suggesting that, the inhibition of a very small proportion of this phosphatase, would be enough to reverse the balance towards mitotic substrate phosphorylation. Alternatively, FAM122A localization could also play an essential role by locally inhibiting the phosphatase at the vicinity of cyclin A/Cdk kinase permitting the phosphorylation of its substrates.

However, strikingly, although FAM122A clearly binds PP2A-B55 during interphase, we discovered that this protein rapidly dissociates from the phosphatase as soon as the extract entered into mitosis and re-associates again upon mitotic exit. Our data show that this dissociation is induced by the phosphorylation of Arpp19 by Gwl. Accordingly, the addition of phospho-S71-Arpp19 to interphase extracts or to FAM122A-PP2A-B55 pull-downs promotes the exchange of FAM122A by phospho-S71-Arpp19 in the PP2A-B55 complex. These data indicate that, when phosphorylated by Gwl, Arpp19 competes with FAM122A and promotes its dissociation from the phosphatase. Indeed, our data demonstrate that phospho-S71-Arpp19 has a 100-fold lower $K_i$ than FAM122A indicating that p-S71-Arpp19 displays much higher affinity, inhibitory activity and binding to PP2A-B55 than FAM122A. Accordingly, in our analysis, endogenous phospho-S71-Arpp19 is able to dissociate ectopic His-FAM122A from PP2A-B55 even when the latter is present in the extract at a molar concentration 215 times higher than the former. These properties were supported by the structure predictions of FAM122A- and Arpp19-PP2A-B55 complexes obtained by AlphaFold2.2. Interestingly, structures predicted for FAM122A-PP2A-B55 and phospho-Arpp19-PP2A-B55 complexes superposed from three different species (human, *Xenopus* and *C. elegans*) revealed that FAM122A and phospho-S71-Arpp19 inhibit the phosphatase by competing with the substrate through directly blocking its access, a hypothesis that is also supported by our kinetics experiments that confirm the competitive model as the best fitting our data. However, FAM122A and phospho-S71-Arpp19 display different interactions with the PP2A-B55 heterocomplex. FAM122A weakly interacts with the C subunit via two glutamic acids forming salt bridges with two arginine residues of the C subunit of PP2A. These interactions locate the second alpha helix of this protein on the surface of the catalytic site occluding, in this way, substrate entry. Conversely, phospho-S71-Arpp19 tightly binds PP2A C mostly because of the presence of the phospho-serine residue that would directly interact with manganese ions of the catalytic site, an interaction that would be stabilized by two supplementary salt bridges formed with two arginine of the C subunit. This tight interaction can be only dissociated upon Arpp19 dephosphorylation, however, as previously reported, dephosphorylation of the Arpp19 Gwl-site is very slow compared to regular substrates of this phosphatase making of this phosphoprotein a potent inhibitor of PP2A-B55 (Labbé et al, 2021; Williams et al, 2014). These data nicely

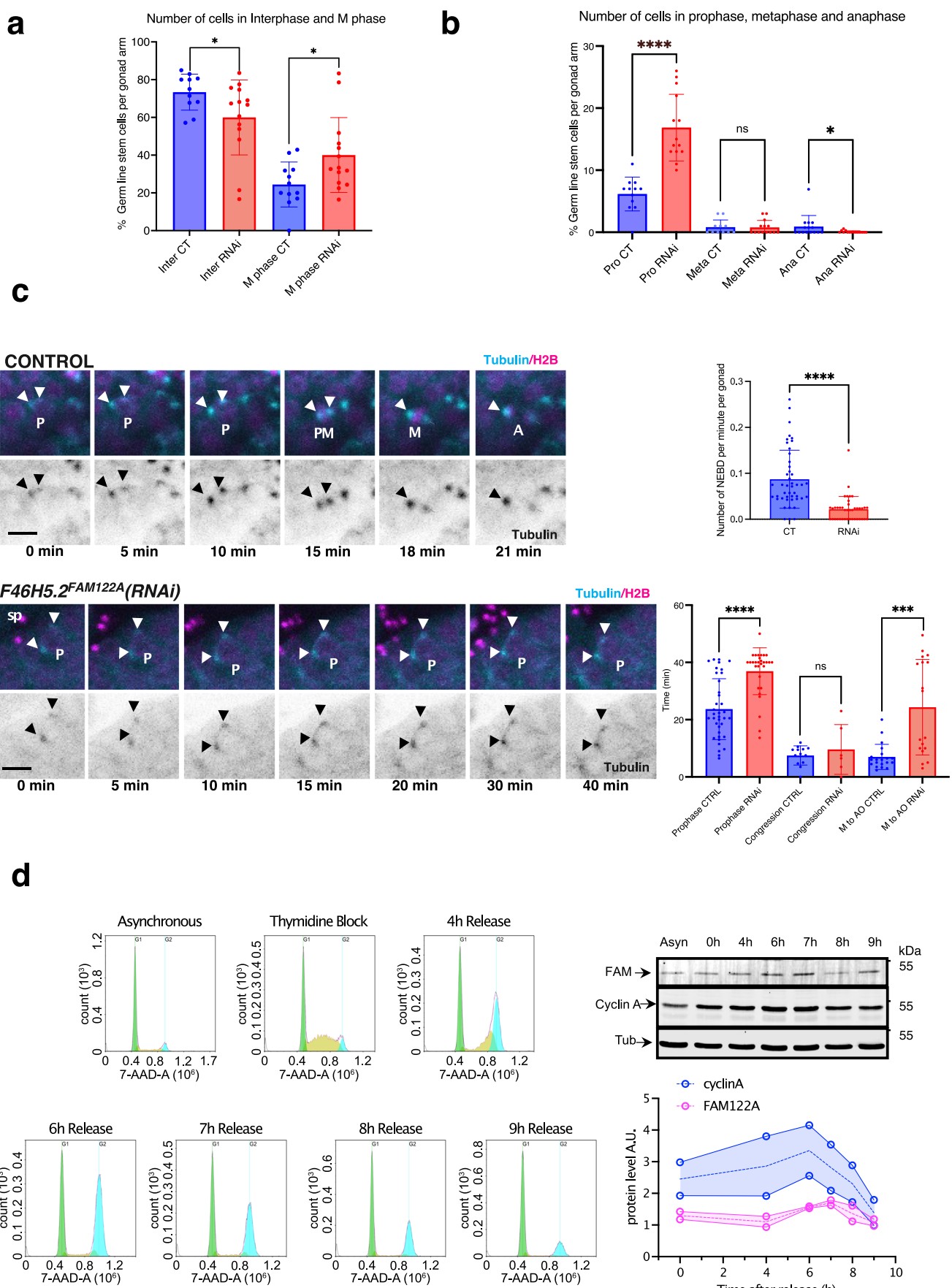

**Figure 8.  FAM122A is required "in vivo" in *C. elegans* to permit mitotic entry and progression in germ stem cells.**

(A) Nematode strains expressing histone and gamma-tubulin tagged with GFP were immobilised and immediately imaged by confocal microscopy for the quantification of interphase and mitotic cells as reported in the scheme of Appendix Fig. S6. Data are represented as mean ± standard deviation. (B) Prophase, metaphase and anaphase germ stem cells in gonads of worms were counted as in (A) and represented as mean ± standard deviation. (C) A GFP-tagged tubulin and with RFP-tagged histone nematode strain was followed by time lapse confocal microscopy and the timing of mitotic progression was determined in germ stem cells. Shown are representative images of stem cells performing mitotic division over time in control and RNAi treated worms. The number of cells performing nuclear envelope breakdown (NEBD) per minute per gonad were counted and represented as the mean values ± standard deviation. Timing of cells to perform prophase (from first centrosome movement to nuclear envelope breakdown), congression (from nuclear envelope breakdown to metaphase plate) or anaphase onset were also recorded and represented as mean values ± standard deviation. Significant differences calculated by non-parametric two tailed Mann–Whitney test are shown. Arrowheads: centrosomes. P:prophase. Sp: Spermatozoids. PM: prometaphase. A: anaphase. Scale bar: 10 µm. *$p < 0.01$; ***$p < 0.001$; ****$p < 0.0001$. $n \geq 10$. (D) RPE1 cells were (Thymidine Block) or not (Asynchronous) synchronized with thymidine treatment and released at the indicated timepoints. Released cells were used for western blot to measure FAM122A, cyclin A and tubulin levels at the different timepoints. Western blot signals were measured and FAM122A and cyclin A amounts were normalized and represented. $n = 2$. Source data are available online for this figure.

explain the different affinities for P22A-B55 of FAM122A and p-S71-Arpp19, however, it is worth highlighting that, the 2 $Mn^{2+}$ divalent cations present in the bacterial purified PP2A-B55 complex used for structural studies, do not reflect the native PP2A-B55 enzyme that contains a $Zn2+: Fe2+$ ion pair (Brautigan and Shenolikar, 2018), a feature that could modify the interaction of p-S71-Arpp19 to the phosphatase. However, during the review process of this manuscript, new cryo-EM structures of the complex of FAM122A or Arpp19 with purified PP2A-B55 from human cells were published (Padi et al, 2024). Interestingly, these structural results confirm our modelling predictions and kinetic data and reinforce the idea that FAM122A and Arpp19 interphases with PP2A-B55 partially overlap. Moreover, according with our model, data also support a role of the phosphorylation of Arpp19 central serine in tightly interacting with the native ions of the catalytic site of the C subunit providing a stronger phosphatase inhibitory activity to this protein compared to FAM122A. However, besides cryo-EM results, authors also presented NMR data suggesting a simultaneously binding of a C-terminal truncated FAM122A and Arpp19 with a loopless B55 form, a B55 variant unable to bind PP2A A subunit. However, whether this trimer can be also observed with the wildtype Arpp19 and FAM122A and the functional PP2A-B55 heterocomplex is not clear. Indeed, these would be very surprising since, as supported by the authors themselves, these two inhibitors interact with the same or very close residues of B55 and C subunits of the phosphatase making highly unlikely the presence of a Arpp19/FAM122A/PP2A-B55 trimer. In this line, our data clearly stablish a dissociation of FAM122A from PP2A-B55 when both proteins are incubated with the phosphatase both in in vitro pulldown experiments and in extracto.

Importantly, our data additionally show that FAM122A is dissociated from PP2A-B55 by phospho-Arpp19 (S71) during mitosis. Although we do not know what could be the role of this dissociation, we predict that it could be physiological relevant and required to induce a correct mitotic progression. In this line, it is possible that a higher PP2A-B55 inhibitory activity is required during mitotic progression to permit the late mitotic substrate phosphorylation. In agreement with this hypothesis, previous data demonstrate that cyclin B1/Cdk1 levels required to enter mitosis are lower than the amount of cyclin B1–Cdk1 needed for mitotic progression (Lindqvist et al, 2007). Moreover, besides the high inhibitory activity of phospho-S71-Arpp19, the fact that its dissociation from PP2A-B55 is gradually induced by dephosphorylation could be also essential to establish the temporal

pattern of mitotic substrate dephosphorylation required for the correct order of mitotic events during mitotic exit.

In summary, we identified a new role of FAM122A in promoting G2-M transition through the inhibition of PP2A-B55 and we provided data indicating that cyclin A/Cdk, together with FAM122A-dependent inhibition of PP2A-B55 promote Gwl and cyclin B/Cdk1 activation and trigger mitotic entry. Finally, we showed that, upon mitotic entry, FAM122A is dissociated from PP2A-B55 by phospho-Arpp19 until the time when Gwl inactivation and Arpp19 becomes fully dephosphorylated permitting to FAM122A to re-associate again to the phosphatase.

## Methods

### "In vitro" phosphorylation

Phosphorylation of Arpp19 on S71 or of *Xenopus* FAM122A by Gwl was induced by using GST-K72M hyperactive mutant form of Gwl purified from SF9 cells. For "in vitro" phosphorylation reaction, *Xenopus* His-Arpp19 protein or His-FAM122A and GST-K72M Gwl kinase were mixed at a final concentration of 133 and 67 µM for the two first proteins and 0.34 µM for the last one in a reaction buffer containing $7 \times 10^{-2}$ µM ATPγ$^{33}$P at a specific activity of 3000 Ci/mmol, 200 µM ATP, 2 mM $MgCl_2$ and Hepes 50 mM, pH 7,5.

For phosphorylation of Arpp19 on S113 by PKA, a final concentration of 33 µM of Arpp19 and 1.25 µM of His-PKA catalytic subunit purified from His-tag column were mixed to the reaction buffer containing $8 \times 10^{-3}$ µM ATPγ$^{33}$P at a specific activity of 3000 Ci/ mmol, 200 µM ATP, 2 mM $MgCl_2$, 100 mM NaCl and 50 mM Tris pH 7.5.

For phosphorylation of GST-PRC1 on T481, Cdk1 activity was obtained by immunoprecipitation. In brief, His-human cyclin A was mixed with CSF extracts at a final concentration of 0.2 µM for 30 min and subsequently supplemented with 50 µl of protein G magnetic Dynabeads pre-linked with 10 µg of *Xenopus* Cdk1 C-terminus antibodies. After 45 min-incubation the beads were washed three times with 500 mM NaCl, 50 mM Tris pH 7.5, twice with 100 mM NaCl, 50 mM Tris pH 7.5 and finally resuspended with 100 µl of reaction buffer containing 1 mM ATP, 2 mM $MgCl_2$, 100 mM NaCl and 50 mM Tris pH 7.5. GST-PRC1 was then added to the beads at a final concentration of 21 µM.

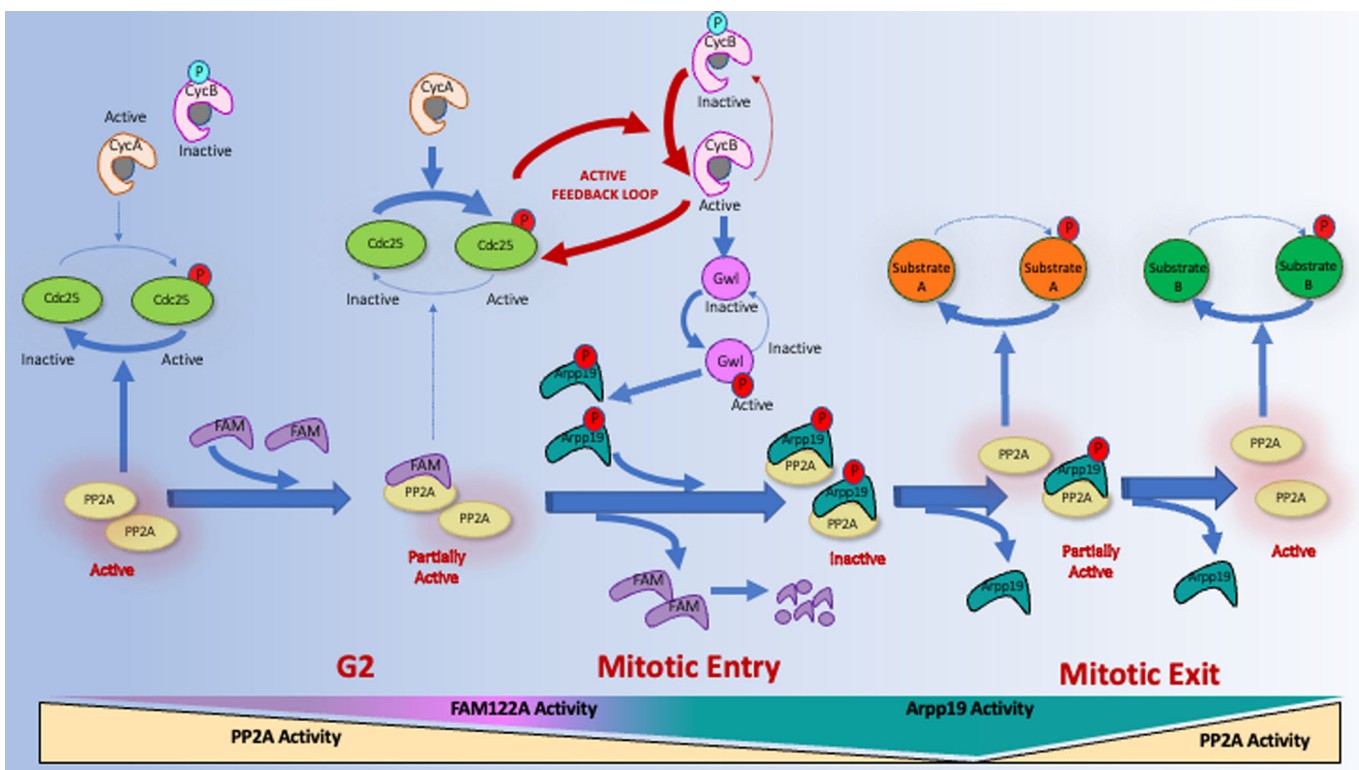

**Figure 9. Hypothetic model representing the network of activation/inactivation of cyclin B/Cdk1 and mitotic entry/exit by the sequential inhibition of PP2A-B55 by FAM122A and Arpp19.**

PP2A-B55 activity is maintained high during G2 preventing the phosphorylation and activation of Cdc25 by cyclin A/Cdk. A first phosphorylation and activation of Cdc25 phosphatase by cyclin A/Cdk is achieved when a critical FAM122A-dependent inhibition of PP2A-B55 is reached. Cdc25 will then dephosphorylate and activate the cyclin B/Cdk1 feedback loop resulting in the activation of Gwl, phosphorylation of Arpp19 and mitotic entry. Phospho-Arpp19 will then induce FAM122A dissociation of PP2A-B55 taking over the inhibition of this phosphatase and promoting the correct temporal dephosphorylation of mitotic substrates. Active and inactive pathways are represented as solid thick arrows and dashed thin arrows respectively. Activatory phosphorylation sites are represented in red whereas inhibitory phosphorylation of Cdk1 is shown in blue.

For thio-phosphorylation of Arpp19 on S71, His-tag of His-Arpp19 protein was removed using TEV protease. Non-tagged Arpp19 was then used for phosphorylation at a final concentration of 66 µM in the presence of 1 mM ATPγS, 2 mM MgCl$_2$, 100 mM NaCl, 50 mM Tris pH 7.5 and 0.34 µM of GST-K72M Gwl kinase.

All "in vitro" phosphorylation reactions were incubated for 1 h, aliquoted and frozen at −70 °C until use.

### Dephosphorylation reactions in kinase inactivated extracts

When the impact of *Xenopus*, human, or *C. elegans* FAM122A proteins on either S113 Arpp19 or T481 PRC1 dephosphorylation was checked in ATP-devoid interphase extracts, 1 µl of the corresponding "in vitro" phosphorylation reaction was mixed to 1 µl His-FAM122A to a final concentration of 14.3 µM, diluted with 8 µl of Tris 50 mM-10 mM EDTA buffer and supplemented with 10 µl of ATP-devoid interphase extracts adjusted to 500 mM NaCl with a solution of 5 M NaCl and a sample of 2 µl was recovered at the indicated time-points.

### Immunoprecipitation/Immunodepletion

Immunodepletions were performed using 20 µl of extracts, 20 µl of G-magnetic Dynabeads (Life Technologies), and 2 µg of each antibody except for cyclin A for which we used 3.3 µl. Antibody-linked beads were washed two times with XB buffer, two times with Tris 50 mM, pH 7.5 and incubated for 15 min at room temperature (RT) with 20 µl of *Xenopus* egg extracts. The supernatant was recovered and used for subsequent experiments. To fully deplete endogenous proteins, two rounds of immunodepletion were performed.

For Gwl rescue experiments, a final concentration of 19.2 nM of K72M GWL was added to depleted egg extracts and a sample of 2 µl was recovered and used for western blot.

For histidine pulldown experiments, 20 µl of interphase or CSF extracts containing His-*Xenopus* or His-human FAM122A proteins, were supplemented with 20 µl of HisPur™NiNTA Magnetic beads (Life Technologies) at a final concentration of 7.15 µM and incubated for 10 min of continuous mixing at 21 °C. Upon centrifugation, beads are washed twice with XB buffer and supplemented with Laemmli sample buffer for western blot use.

## Protein purification

His-*Xenopus* Arpp19, His-*Xenopus* wildtype and mutant of FAM122A in which all serine and threonine residues were mutated into alanine, as well as, His-human FAM122A, His-*C elegans* FAM122A, His-human PRC1, and His-Rat Catalytic Subunit of PKA were produced in *Escherichia coli* and purified using TALON Superflow Metal Affinity Resin. GST-*Xenopus* FAM122A protein was produced in *Escherichia coli* and purified using a glutathione column.

## CSF, interphase and kinase-inactivated egg extracts

Frogs were obtained from ≪ TEFOR Paris-Saclay, CNRS UMS2010 / INRAE UMS1451, Université Paris-Saclay≫, France and kept in a *Xenopus* research facility at the CRBM (Facility Centre approved by the French Government. Approval no. C3417239). Females were injected of 500 U Chorulon (Human Chorionic Gonadotrophin) and 18 h later laid oocytes were used for experiments. Adult females were exclusively used to obtain eggs. All procedures were approved by the Direction Generale de la Recherche et Innovation, Ministère de L'Enseignement Supérieur de la l'Innovation of France (Approval no. APAFIS#40182-202301031124273v4).

Metaphase II-arrested egg extracts (CSF extracts) were obtained by crushing metaphase II-arrested oocytes in the presence of EGTA at a final concentration of 5 mM (Lorca et al, 2010).

Interphase *Xenopus* egg extracts were obtained from metaphase II-arrested oocytes 35 min after $Ca^{2+}$ Ionophore (final concentration 2 µg/ml) treatment.

To measure the capacity of FAM122A to promote mitotic entry, 20 µl of CSF or Interphase extracts were supplemented with a final concentration of 7.15 µM of *Xenopus*, human or *C elegans* FAM122A proteins.

For dephosphorylation assays kinase inactivated interphase extracts were mixed with Arpp19 or PRC1 proteins "in vitro" phosphorylated by PKA or by cyclin A/Cdk1 respectively at a final concentration of 1.65 µM together with *Xenopus* or human FAM122A at a final concentration of 14.3 µM.

Kinase-inactivated egg extracts were obtained from dejellied unfertilised eggs transferred into MMR solution (25 mM NaCl, 0.5 mM KCl, 0.25 MgCl$_2$, 0.025 mM NaEGTA, 1.25 mM HEPES-NaOH pH7.7), washed twice with XB Buffer (50 mM sucrose, 0.1 mM CaCl$_2$, 1 mM MgCl$_2$, 100 mM KCl, HEPES pH 7.8) and centrifuged twice for 20 min at $10,000 \times g$. Cytoplasmic fractions were then recovered, supplemented with RNAse (10 µg/ml final concentration) and dialyzed versus a solution of 50 mM Tris pH 7.7, 100 mM NaCl overnight to eliminate ATP. Upon dialysis, extracts were ultracentrifuged for 50' at $300,000 \times g$ and supernatant recovered for use.

## K$_i$ values for FAM122A and p-S71-Arpp19 inhibitors

Arpp19 phosphorylated on S113 by PKA was mixed with kinase inactivated interphase extracts at final concentrations of 120, 180, 300, 600, 1200 and 2000 nM. Since at lower concentrations of p-S113-Arpp19, the substrate is more rapidly dephosphorylated, a dilution of the PP2A-B55 phosphatase was required to record sufficient dephosphorylation experimental points on initial linear phases. We thus diluted the extracts from 1/50 to 1/200 when the

concentration of p-S113-Arpp19 varied from 120 to 2000 nM. Upon substrate addition to the extracts, an aliquot was taken every one or two minutes until 7 or 14 min respectively, depending on the substrate concentration. The reaction was stopped by adding Laemmli blue buffer and heating for 5 min at 90 °C.

Dephosphorylation was measured by western blot using anti-pS113-Arpp19 antibody, a secondary goat anti-rabbit DyLight 800 conjugated antibody and quantified by Li-cor Odysey M System. Inhibition constants (K$_i$) were determined by repeating these experiments at different concentrations of FAM122A (50, 125 and 250 nM) or Thio-S71-Arpp19 (1, 2 and 3 nM). Values of the steady state rate constants, $k_{ss}$, were determined using GraFit 7.0.3 software (Erithacus software) from the slope of the initial linear phases of the reaction time courses. The actual sampling times were normalized by dividing by the enzyme dilution factor, so that all experiments could be combined for global fitting procedures. Global fittings were performed on the raw data with GraFit 7.0.3 software using four different inhibition modes: competitive, non-competitive, mixed and uncompetitive. Times courses and secondary curves and the corresponding Lineweaver-Burk representations are shown as Appendix Materials (Appendix Figs. S3, S4).

## Plasmids

*Xenopus* wildtype and the Δ(1–73) and serine/threonine-to-alanine mutant form (accession number NP_001085566.1) cDNAs, as well as human (accession number NP_612206.5), and *C elegans* (accession number NP_001024675.1) cDNAs were synthesized by GeneCust (France) and subcloned into the HindIII-XhoI site of pET14b for human FAM122A, into the NdeI-BamHI site of pET14b for *C elegans*, into the XhoI-BamHI site of pET14b for *Xenopus* wildtype and mutant forms and into the BamHI-XhoI site of pGEX4T1 for wildtype *Xenopus* FAM122A.

## Antibodies

*Xenopus*, human and *C elegans* FAM122A protein was detected using anti-histidine antibodies. Antibodies used in this study are detailed in Appendix Table S1.

## Mutagenesis

Deletions and single-point mutations of *Xenopus* FAM122A were performed using Pfu ultra II fusion DNA polymerase. Oligonucleotides were purchased from Eurogentec and are detailed in the Appendix Table S2.

## *C. elegans* culture and RNAi mediated depletion

*C. elegans* worm strains N2 (wildtype ancestral, Bristol) and MT2124 (*let-60(gf)*) were obtained from the CGC (https://cgc.umn.edu) and maintained at 20 °C on NGM plate using standard procedures (Brenner, 1974) except that worms were fed with HT115 bacteria to standardize their growth condition with the RNAi mediated depletion. RNAi feeding was performed as described previously (Kamath et al, 2001). HT115 thermocompetent *E. coli* were transformed with L4440 empty vector (control) or containing a sequence targeting, SUR-6$^{PP2AB}$ or F46H5.2$^{FAM122A}$. The

L4440 vector targeting F46H5.2[FAM122A] was generated using a fragment amplified from *C. elegans* cDNA corresponding to 715 bp of the coding sequence (nucleotides 398 to 1112) which was subsequently inserted between XhoI and NotI restriction enzyme sites in L4440 vector. Control, SUR-6[PP2AB] vector come from the Arhinger's library. In a 14 ml culture tube, 2 ml of LB medium supplemented with 100 µg/mL of ampicillin were inoculated with a colony of HT115 bacteria transformed with respective L4440 vectors and incubated at 37 °C under agitation. After 7 h, 200 µl of this bacterial culture was transferred on a 60 mm diameter NGM plate containing 0.2 mM IPTG and 50 µg/ml of carbenicillin. Plates were allowed to dry overnight at room temperature, stored at 4 °C and used within 48 h. Worms were synchronized using the alkaline bleach method (1.2% NaOCl, 250 mM KOH in water (Stiernagle, 2006)). Eggs obtained from the alkaline bleach were allowed to hatch overnight at 16 °C in M9 buffer. To score FAM122A effect on Ras (*let-60(gf)*) induced multivulva phenotype, around 200 synchronized larvae were placed on a single feeding plate and incubated 66–72 h at 20 °C until control worms reach adulthood. Phenotypes were scored at the time where the first eggs laid by control worms started to hatch to ensure that all worms fully develop to the adult stage. Vulval defect phenotypes was scored under a dissecting scope by counting the total number of worms and the number of worms exhibiting multiple vulva. The counting was repeated once for each plate to minimize scoring errors. Representative images of vulval phenotypes were acquired with a scMOS ZYLA 4.2 M camera on a Zeiss Axioimager Z2 and a 20X Plan Apochromat 0.8 NA using worms immobilized with a solution of 10 mM sodium azide and mounted on an 3% agarose pad in between a microscopy slide and a cover glass.

## Live imaging of *C. elegans*

For "in vivo" imaging, we used nematode strains expressing GFP-gamma-tubulin and GFP-histone (TH32: unc-119(ed3), ruIs32[-pAZ132; pie-1/GFP::histone H2B] III; ddIs6[GFP::tbg-1; unc-119(+)] V or GFP-tubulin and histone-mCherry (JDU19: *ijmSi7 [pJD348; Pmex-5_gfp::tbb-2; mCherry::his-11; cb-unc-119(+)] I; unc-119(ed3) III)* and JDU233: *ijmSi63 [pJD520; mosII_5'mex-5_GFP::tba-2; mCherry::his-11; cb-unc-119(+)] II; unc-119(ed3) III)* kindly provided by Julien Dumont (Institut Jacques Monod, Paris). Worms were cultured at 25 °C and bacterial feeding was performed similarly to the multivulva experiment, except that worms were not synchronized by alkaline bleach. Instead, 6 worms were allowed to lay eggs on a 60 mm diameter NGM feeding plate. After 2 h at 25 °C adults were removed and their progeny was allowed to develop on the feeding plate and at 25 °C for 44–52 h until the first adults start to lay eggs (young adults). Worms were anesthetized in 0.02% tetramisole in M9 buffer for 10 min before being transferred onto a 4% agarose pad (Laband et al, 2018). The pad was then covered with 22 × 22 mm coverslip and sealed with VaLaP (vaseline and paraffin wax and lanolin, 1:1:1). The chamber was filled with M9 to prevent drying and to dilute tetramisole. Immobilized worms were imaged for a maximal time of 50 min and were not maintained more than 1 h after being removed from their feeding plate. Germline stem cell division imaging was performed according to Gerhold et al, (Gerhold et al, 2015). To minimize toxicity, we used minimal laser intensity and exposure using a spinning disk confocal Yokogawa W1 on an Olympus inverted microscope coupled to a sCMOS Fusion BT Hamamatsu camera. Long-term live imaging (up to 50 min) was performed with a 30× UPLSAPO 1.05 NA DT 0.73 mm silicone objective by taking a full z-stack of the entire worms (~40 µm) with 2 µm steps every 20 s or 30 s. For higher resolution images and to score mitotic phases using centrosome and chromatin markers, a single z-stack was performed on each worm with a 1 µm z-section and using a 60× UPLSAPO 1.3 NA DT 0.3 mm silicone objective.

## Analysis of germline stem cells divisions

Images and movies were visualized using ImageJ or CellSens visualization software (OlyVIA, Olympus). Analysis of germline stem cells divisions was made following a previous article from Gerhold et al (Gerhold et al, 2015). Accordingly, timing of prophase corresponds to the initiation of centrosome separation (GFP-tubulin) to the beginning of chromosome congression monitored using the mCherry-histone. As ovulation generates movement within the gonad arm, cells can move in 3D, therefore tracking of individual cell progression was performed by navigating throughout z-stack over time. Representative images and montages in Fig. 8 correspond to a maximal intensity projection of 4–5 z-sections (8–10 µm). To quantify the number of cells in each respective cell cycle phases, we used GFP-histone and GFP-gamma-tubulin signal to monitor centrosome separation, nuclear envelope breakdown, chromosome congression and separation. The number of stem cells in the mitotic zone of the distal gonad was estimated by counting the number of nuclei based on histone signal. Here, cells were considered in prophase when having two separating or opposed centrosomes (spots) closed to nuclei or condensed chromosomes. Metaphase and anaphase correspond respectively to a single plate or two separated plates of tightly condensed chromosomes.

## Protein structure modelling

Complexes of PP2A-B55 with Arpp19/ENSA or FAM122A were produced using Alphafold_Multimer version 2.2 (preprint: Evans et al, 2021) with the protein sequences from three species (*C. elegans, X. laevis* and *H. sapiens*). Manganese ions were added to the resulting models by superposition and extraction from the crystal structure of PP2A (PDB3DW8). The phosphoserine was modelled by superposing a phosphoserine onto the corresponding serine in each complex of interest and searching for the best orientation using our recent webserver ISPTM2 (https://isptm2.cbs.cnrs.fr). Because full-length FAM122A sequences led to no interactions as predicted by Alphafold 2.2, truncated version of FAM122A were used instead. For the human, *Xenopus*, and worm sequences, the shortened versions comprised residues 84–117, 61–120 and 70–149, respectively.

## Cell culture and synchronization

RPE1 cells were maintained adherent using standard cell culture procedures in DMEM-F12 medium supplemented with 10 mM HEPES pH 7.2, 10% of fetal bovine serum and antibiotics (penicillin, streptomycin). Synchronization of cells at the G1/S boundary was achieved by a double thymidine block. In brief, cells were plated at 14,000 cells per $cm^2$ in 6 cm diameter petri dishes

and, 24 h later, incubated for 18 h in medium containing 2 mM thymidine. Cells were then washed twice with 1xPBS to remove excess of thymidine and fresh medium containing 25 μM of 2'-deoxycytidine was added for 8–9 h. Cells were washed twice with 1xPBS and a second block was performed by adding medium containing 2 mM of thymidine for 18 h. Release of G1/S arrested cells was performed by washing twice with 1× PBS and exchanging the medium for fresh medium containing 25 μM of 2'-deoxycytidine. Cells were collected at 0, 4, 6, 7, 8 and 9 h after release. Cells were collected and analysed by western blot analysis and flow cytometry (FACS). For western blot analysis, 40 μg of total protein were loaded and separated by SDS-PAGE using 10% acrylamide gels. Proteins were then transferred onto a PVDF membrane compatible with fluorescent analysis. Protein amounts were measured by western blot using polyclonal anti-FAM122A (1/1000) and anti-cyclin A (1/2000) and monoclonal anti-βTubulin (1/2000) primary antibodies and with goat anti-rabbit Dylight 680 conjugated and goat anti-mouse Dylight 800 conjugated fluorescent secondary antibodies and quantified by Li-cor Odyssey M System. For cell sorting analysis, after ice-cold 70% ethanol fixation at −20 °C, cells were resuspended and incubated for 5 min in 500 μL of 0.4%TritonTX100, 3% BSA in 1× PBS for cell permeabilization. DNA staining was then performed by adding 300 μL of 1xPBS containing 3% BSA, 0,01 μg/mL of 7-AAD and 0.01 mg/mL RNAse A. Cells were incubated in the dark for 1 h at room temperature, washed two times with 1× PBS supplemented with 3% BSA by centrifugation (3 min at 400 × g) and resuspend in 130 μL prior to cell sorting using Novocyte flow cytometer system.

## Data availability

Live imaging data sets used in this manuscript are available on BioImage Archive with the accession number: S-BIAD984.

## Peer review information

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

## Acknowledgements

We are grateful to Marc Plays and Phillipe Richard for animal and antibody production and to MRI for microscopy facility. We thank Julien Dumont (Institut Jacques Monod, Paris) for kindly providing GFP-gamma-tubulin and GFP-histone nematode strains and Lucie Van Hove, Sylvain Roque, Morgane Robert, Lucie Van Hove (LP lab) and Celia Benchoug for their technical help. This work was supported by the Agence National de la Recherche (REPLIGREAT, ANR-18-CE13-0018-01, MITODISSECT, ANR-22-0022 and MTDiSco, ANR-20-CE13-0033), La Ligue Nationale Contre le Cancer (Equipe Labellisée, EL2019 CASTRO and EL2018 PINTARD), Ligue Nationale Contre le Cancer (Comité Département 66/ LACROIX), the French Infrastructure for Integrated Structural Biology (FRISBI, ANR-10-INSB-005 and the infrastructure ChemBioFrance. Some nematode strains were provided by the CGC, which is funded by NIH Office of Research Infrastructure Programs (P40 OD010440).

## Author contributions

**Benjamin Lacroix**: Conceptualization; Software; Investigation; Methodology. **Suzanne Vigneron**: Investigation; Methodology. **Jean Claude Labbé**: Investigation. **Lionel Pintard**: Methodology; Resources. **Corinne Lionne**: Investigation; Methodology. **Gilles Labesse**: Software; Methodology. **Anna Castro**: Software; Formal analysis; Funding acquisition; Validation; Writing—original draft; Writing—review and editing. **Thierry Lorca**: Conceptualization; Formal analysis; Supervision; Funding acquisition; Validation; Investigation; Methodology; Writing—original draft; Writing—review and editing.

## Disclosure and competing interests statement

The authors declare no competing interests.

