## [Peer Review File · The EMBO Journal]

Increases in cyclin A/Cdk activity and in PP2A-B55 inhibition by FAM122A are key mitosis-inducing events

Benjamin Lacroix , Suzanne Vigneron , Jean Claude Labbé , Dr. Lionel Pintard , Corinne Lionne , Dr. Gilles Labesse , Dr. Anna Castro

Corresponding author(s): Thierry Lorca, Anna Castro

Review Timeline:

Submission Date:	19th Jun 23
Editorial Decision:	28th Jul 23
Revision Received:	22nd Dec 23
Editorial Decision:	22nd Jan 24
Revision Received:	31st Jan 24
Accepted:	6th Feb 24

Editor: Hartmut Vodermaier

Transaction Report:

Dr. Thierry Lorca
CNRS
Cell Cycle
1919 route de Mende
Montpellier 34293
France

28th Jul 2023

Re: EMBOJ-2023-114793
Cyclin A/cdk activity and FAM122A-dependent inhibition of PP2A-B55 are essential to trigger mitosis.

Dear Anna and Thierry,

Thank you again for submitting your study on FAM122A roles in triggering mitotic entry to The EMBO Journal, and apologies for the delay in getting back to you with a decision. We have now finally received a complete set of reviews from three expert referees, copied below for your information. As you will see, all referees acknowledge the importance of the question and the potential interest of your new results, but also raise a number of well-taken questions that would need to be satisfactorily answered prior to acceptance. Should you be able to adequately address these concerns, we would be happy to consider a revised manuscript further for publication.

Since it is our policy to consider only a single round of major revision and therefore important to fully answer to all comments at the time of resubmission, I would invite you to get back to me with a tentative response letter/revision plan already during the early stages of the revision work. On the basis of this response, we could then further discuss how certain conceptual points, especially those raised by referees 1 and 2, might be clarified or answered, and which experimental extensions would be required/which might be dispensable for a successful revision. After my return from vacation in the second half of August, I would be happy to schedule a video call for such discussions based on such a tentative response. I should add that we could also offer extension of the default three-months revision period if needed, with our 'scooping protection' (meaning that competing work appearing elsewhere in the meantime will not affect our considerations of your study) remaining of course valid also throughout this extension.

Detailed information on preparing, formatting and uploading a revised manuscript can be found below and in our Guide to Authors. Thank you again for the opportunity to consider this work for The EMBO Journal, and I look forward to hearing from you in due time.

With kind regards,

Hartmut

*** PLEASE NOTE: All revised manuscripts are subject to initial checks for completeness and adherence to our formatting guidelines. Revisions may be returned to the authors and delayed in their editorial re-evaluation if they fail to comply to the following requirements (see also our Guide to Authors for further information):

- 3) Revised manuscript text (including main tables, and figure legends for main and EV figures) has to be submitted as editable text file (e.g., .docx format). We encourage highlighting of changes (e.g., via text color) for the referees' reference.
- 4) Each main and each Expanded View (EV) figure should be uploaded as individual production-quality files (preferably in .eps, .tif, .jpg formats). For suggestions on figure preparation/layout, please refer to our Figure Preparation Guidelines: <http://bit.ly/EMBOPressFigurePreparationGuideline>
- 5) Point-by-point response letters should include the original referee comments in full together with your detailed responses to them (and to specific editor requests if applicable), and also be uploaded as editable (e.g., .docx) text files.
- 6) Please complete our Author Checklist, and make sure that information entered into the checklist is also reflected in the manuscript; the checklist will be available to readers as part of the Review Process File. A download link is found at the top of our Guide to Authors: embopress.org/page/journal/14602075/authorguide
- 7) All authors listed as (co-)corresponding need to deposit, in their respective author profiles in our submission system, a unique ORCID identifier linked to their name. Please see our Guide to Authors for detailed instructions.
- 8) Please note that supplementary information at EMBO Press has been superseded by the 'Expanded View' for inclusion of additional figures, tables, movies or datasets; with up to five EV Figures being typeset and directly accessible in the HTML version of the article. For details and guidance, please refer to: embopress.org/page/journal/14602075/authorguide#expandedview
- 9) Digital image enhancement is acceptable practice, as long as it accurately represents the original data and conforms to community standards. If a figure has been subjected to significant electronic manipulation, this must be clearly noted in the figure legend and/or the 'Materials and Methods' section. The editors reserve the right to request original versions of figures and the original images that were used to assemble the figure. Finally, we generally encourage uploading of numerical as well as gel/blot image source data; for details see: embopress.org/page/journal/14602075/authorguide#sourcedata

At EMBO Press, we ask authors to provide source data for the main manuscript figures. Our source data coordinator will contact you to discuss which figure panels we would need source data for and will also provide you with helpful tips on how to upload and organize the files.

In the interest of ensuring the conceptual advance provided by the work, we recommend submitting a revision within 3 months (26th Oct 2023). Please discuss the revision progress ahead of this time with the editor if you require more time to complete the revisions. Use the link below to submit your revision:

Link Not Available

Referee #1:

This is a nice paper describing a series of elegant experiments that establish that the FAM122A proteins acts transiently as an inhibitor of PP2A-B55 protein phosphatase catalytic activity at the beginning of mitosis, before being displaced from PP2A-B55 by phosphoARRP19, which following phosphorylation by the Greatwall mitotic protein kinase binds more tightly to the PP2A C subunit than FAM122A, whose binding is phosphorylation independent. This initial inactivation of PP2A-B55 upon entry into mitosis allows cyclinA/Cdk phosphorylated substrates to accumulate leading to activation of cyclin B/Cdk1 and Greatwall, which triggers ARPP19/ENSA phosphorylation resulting in more stable inhibition of PP2A-B55 and mitotic progression.

Points: 1. Figure 1A/C: It is obvious that the added FAM122A was phosphorylated in the extracts with kinase activity, but the authors did not discuss this or the possible importance of these phosphorylation in FAM122A function. There are a number of S/T.P sites in FAM122A. Is FAM122A a substrate for PP2A-B55

2. Figure 1B: How does level of FAM122A protein added to the Xenopus extract relate to the level of endogenous Xenopus FAM122A?

3. Can the authors use BLI or some other biophysical assay to measure the actual affinity of FAM122A binding to PP2A-B55? Does a S62A mutant ARPP19 not have any affinity for PP2A-B55 - in their structural model in Figure 6 there are other protein-protein contacts between ARPP19 and PP2A in addition to the phosphate interactions.
4. Figure 2A: Since the blot in the sixth panel was done with anti-pY15 Cdk1 antibodies, I recommend that all the panels of this sort in the paper be labeled anti-pY15 rather than P_{Tyr}.
5. Figure 2F: Again, some comment on the heavy phosphorylation of FAM122A, especially when cyclin A was added in the right hand panel would be helpful. How many phosphosites are there in FAM122A, and could any of them be relevant to FAM122A/PP2A-B55 interaction?
6. Figure 3A: A linear schematic of the FAM122A protein might also be helpful
7. Figure 3E: What is the lower FAM122A band in the WT lane - is it a cleavage fragment? What antibody was used for blotting in this panel?
8. Figure 3F: These mutation data look convincing - can the essential helical residues be displayed on a helical wheel? Can binding of an isolated H1 peptide to B55 be detected? Likewise, does an H2 peptide alone bind/inhibit PP2Ac?
9. Figure 5: Different and very precise amounts of Xe FAM122, Thio-Arpp19, His-Arpp19, and GST-Xe FAM122A proteins were added in these experiments, but the rationale for using these amounts was not explained, and it is unclear how the of the final concentrations achieved by addition of these recombinant proteins related to the endogenous levels of ARPP19 and FAM122A.
10. Figure 5B: The authors should note that GST-FAM122A is dimeric and this will have a significantly higher avidity for PP2A-B55 than monomeric FAM122A, which might affect the interpretation of these experiments.
11. Figure 5C: This is not described in the text very well, i.e. the fact that this depends upon dephosphorylation of pS67 was not made explicit in the relevant paragraph on page 9. .
12. Figure 6B: The pSer62 (human) and pSerS71 (Xenopus) and pSer61 (C. elegans) residue should be labeled pS62, pS71, pS61 and the phosphate in this residue should be depicted in a different color than amber so it stands out for the reader. Obviously, it would be nice to have real crystal structures of these complexes to confirm the AlphaFold2-multimer predictions, but this would be beyond the scope of this paper.

Minor point: The extensive summary of all the results in the final long paragraph of the Introduction does not really belong here but rather at the start of the Discussion.

Referee #2:

My apologies for the short delay in returning this review. I somehow thought I had more time than was requested, and I became sick in the interim.

Overall, I strongly support the publication of this manuscript in The EMBO Journal, as I think the findings are significant and for the most part well documented. However, some aspects of the manuscript were problematic and call for at least for some rewriting; whether some additional experiments should be required I leave to the Editor's discretion.

The strengths of the paper include a variety of approaches that together establish a coherent and plausible case for the function of FAM122A inhibition of PP2A-B55 in mitotic entry and its displacement from PP2A-B55 during M phase because it is outcompeted by pArpp19/Endosulfine. These approaches include the use of Xenopus egg extracts, structural studies based on AlphaFold, mutagenesis of key sites in FAM122A, and an RNAi/cell biological experiment in C. elegans showing the in vivo significance of these novel findings.

One major concern is inherent in the title, which claims that increases in both the level of CyclinA/cdk and FAM122A-dependent inhibition of PP2A-B55 are the key events triggering mitotic entry. There is no question that CyclinA/cdk is part of this trigger, and that FAM122A inhibition of PP2A-B55 is of importance to the system; this latter point is in fact in my mind the key contribution of this manuscript. However, the paper provides no evidence whatsoever that FAM122 inhibition increases prior to M phase entry. In line 86, the authors state that FAM122A accumulation during G2 could be the first events triggering mitotic entry. But the manuscript provides no evidence for such an accumulation, nor are any references cited. The authors themselves show nicely that FAM122A phosphorylation does not affect its inhibitory activity, so post-translational modifications appear not to be involved. It is possible that the nuclear/cytoplasmic localization of FAM122A changes at this time, a point which has been claimed in some but not all previous publications, but the authors do not deal with this issue and it would be a moot point with respect to Xenopus extracts. Thus, this part of the conclusion is oversold. If FAM122A activity does not change at the G2-M

transition, it is misleading to call it part of the trigger, at least in the biological sense of being some change in state leading to something else.

This being said, I think the demonstration that FAM122A is important for progression into M phase, most convincingly in the *C. elegans* RNAi experiment, is nonetheless important and worthy of publication in *The EMBO Journal*. Some of the importance lies in some history of which the authors do not seem to be aware. The existence of strong overall PP2A-B55 inhibition during interphase was previously suggested by simple *Xenopus* extract dilution experiments done by Mochida and Hunt and later by Williams et al. Simply put, the activity of PP2A-B55 increases strongly as interphase extracts are depleted; this is due to a decrease in the concentration of inhibitors in the extract. The degree of inhibition in an extract is inversely proportional to $(1 + \text{the sum for each inhibitor of the ratio of the concentration of inhibitor to the } K_i \text{ of the inhibitor})$. Note that in this sense all substrates of the enzyme are inhibitors of other substrates; in such cases the ratio is the concentration of substrate to the K_m . Based on the results of Mochida and Williams, this sum for PP2A-B55 in interphase extracts is ~50. What this means is that only about 2% of PP2A-B55 enzymes are available for working on any new substrate molecule. And this is before Gwl phosphorylation of Arpp19/Endosulfine creates an even stronger inhibitor whose ratio in a mitotic extract is between 1000-2500 (because it is a much stronger inhibitor).

The question is: what fraction of "50" is due to FAM122A? I had previously thought that most of this number could be accounted for by the many substrates of PP2A-B55, which must collectively be at high concentrations even if the K_m s were high. However, this manuscript shows that my supposition is wrong and some substantial fraction of "50" must be due to FAM122A, but I don't know what that fraction is. I regard the answer to this question as important because it would provide some guidance as to the degree to which FAM122A inhibits PP2A-B55 in the cell. If this number is low, then FAM122A likely works on only a subset of possible substrates that it can outcompete, or it works in a specific cellular compartment. If the number is high (closer to 50), this would raise the question of why cells make so much PP2A-B55 if 98% of the enzyme is always occupied by FAM122A (unless this changes at some point in the cell cycle, an issue that seems to remain unresolved).

The data in the paper provide some tantalizing but ultimately frustrating clues about the relevant numbers. Endogenous pArpp19 can displace 200X more FAM122A from PP2A-B55; given that the K_m for pArpp19 is about 1 nM, this might suggest that the K_i for FAM122A is minimally say 100 nM. For FAM122A to account for all of the "50", then its endogenous concentration would need to be 5 microM, which is almost certainly much too high. Another possible calculation is based on the data in Figure 1 suggesting that the addition of (what I think is) ~4 microM FAM122A reduces the rate of substrate dephosphorylation by PP2A-B55 about 10-fold, which suggests that the endogenous FAM122A might be about 400 nM (requiring the K_i to be 10 nM, which seems unlikely given the displacement experiment).

Clearly, I don't know the answers here, but the paper would be substantially improved with some additional attention to quantitation. One aspect of this is that the figure legends should state concentrations of at least the components that are added, instead of amounts or volumes. There are many other strategies that could be employed to estimate FAM122A concentrations or its K_i . The authors apparently made antibody to this protein. Unfortunately it was not usable for immunoprecipitation, but perhaps it is good enough for Western blotting, in which case the endogenous concentration of FAM122A could be measured. Additional experiments such as those in Fig. 1, with more early timepoints and different concentrations of added FAM122A could help determine the K_i . Dilution experiments like those done by Mochida could be used to estimate the ratio of concentration to K_i . Perhaps the K_i could be estimated by the contact area predicted by AlphaFold. Etc.

Another semi-major issue concerns the authors' repeated supposition that the divalent cations in the PP2A catalytic subunit are Mn^{2+} . This is true of the enzyme synthesized in bacteria, but it is not true of the native PP2A enzyme, where the divalent cations are Fe^{2+} and Zn^{2+} . (See Bruatigan and Shenolikar 2018 *Ann. Rev. of Biochem.*). The difference is huge from the enzyme kinetic perspective: reaction rates and K_m 's increase 2-3 orders of magnitude when the divalent cations are Mn^{2+} instead of Fe^{2+} and Zn^{2+} , and the enzyme has other artifactual properties. I doubt that this difference is relevant in terms of the structure of the complex of PP2A-B55 with FAM122A, because the inhibitor does not seem to make close contact with the active site, and also because the identity of the divalent cations does not seem to change the 3D structure (see Brautigam and Shenolikar). However, the complex with pArpp19 is a different story: the identity of the divalent cations is key to the mechanism. In lines 298-299, the authors say that the present modeling does not permit them to understand the slow dephosphorylation of pArpp19. This might be true of the Mn^{2+} enzyme, but it is clear that in the native enzyme, the contacts of the phosphoserine with the divalent cations are very close, making a stable low energy intermediate. It is this stability that dictates the high affinity of pArpp19 for the active site and also its slow dephosphorylation.

As stated at the outset, I do think this manuscript is novel and significant and should be published in the *EMBO Journal*. However, I do have the following concrete recommendations for the writing:

(1) Unless the authors have evidence or other justification for the existence of changes in FAM122A activity at the G2-M transition, the title of the paper needs to be changed, and less emphasis placed in the Introduction and Discussion about it being part of a trigger. What remains is still important and justifies publication: FAM122A establishes a precondition needed for cyclinA/cdk to initiate M phase by inhibiting some significant (but currently unknown) fraction of PP2A-B55.

(2) The authors should at a minimum clarify that the Mn^{2+} PP2A structure is of artifactual, bacterially produced enzyme, that the

native enzyme has other divalent cations, and that this fact could significantly alter the way in which pArpp19 interacts with the active site.

(3) The figure legends should be redone with statements of reagent molarities (rather than amounts or volumes) where appropriate. It is often impossible to follow what they did with various dilutions and mixings, and readers should not be forced to calculate molarities themselves.

(4) It was annoying that the authors rewrote history (in line 298 and 461) that the study establishing pArpp19/Endosulfine works as an inhibitor of PP2A-B55 that is slowly dephosphorylated was their reference published in 2021, when this principle was already established in 2014 (their reference 21).

(4) The paper could use some careful editing. Much of the paper is well-written, but here and there sentences crop up with incorrect English constructions that are confusing. I noticed at least one bad typo, in which the concentration of endogenous Arpp19 is stated in line 255 to be 0.115 microM, whereas in Supplementary Figure 2 it is 115 microM. The manuscript is so dense that I gave up trying to document these issues.

The manuscript would be improved with some additional experimentation, but given the already substantial work put into the paper, I am agnostic as to whether some of the following experiments should be required prior to publication.

(1) As outlined above, several approaches exist that could clarify the concentration of FAM122A and also the Ki. Some of these approaches are in theory fairly simple: for example, if the authors have antibody that detects FAM122A on Western blots (even if it does not work for immunodepletion), it should be fairly simple to determine the endogenous amounts. Perhaps I am in the minority by being interested in these numbers, but I believe it would increase our understanding of the role of this protein to get some idea of what fraction of PP2A-B55 it is likely to inhibit.

(2) It is not clear to me if AlphaFold is sufficiently accurate and robust so that substitution of the divalent cations in the structure of the complex with pArpp19 would show anything not visible in the current model. I also don't know how much computational effort would be involved in doing this substitution. What I do know is that the right divalent cations in the active site do in fact make a huge difference.

Referee #3:

This is an interesting manuscript, identifying that FAM122A is important for mitosis by regulating phosphatase activity. The authors show that addition of FAM122A can bring xenopus extracts to mitosis, and that FAM122A interacts with PP2A. They further show a hand-over mechanism, after greatwall activation inhibition of PP2A is performed by ARPP19/ENSA. Depletion in xenopus unfortunately does not work, but instead the effect is tested in *c elegans*. There a setup shows that FAM122A can counteract PP2A, and that depletion gives more mitotic cells and longer time with separated centrosomes.

In general I'm very positive to this manuscript and think it can be an important contribution to the field, however, I have some points that I think should be addressed before publication.

There are hard statements that FAM122A works as an initial trigger for mitosis. This is based on hard statements regarding that phosphorylation of FAM122A does not impact its properties as a mitotic inducer when added to extracts. I have two major problems with this. First, it is unclear to me how much FAM122A is added to extracts in relation to the endogenous protein and whether the setup could detect modification of activity. Key experiments in Figure 2 should be reperformed using a titration to test whether the read-out is dose dependent and to assess whether the S/T mutant can affect activity. Second, phosphorylation of FAM122A has been suggested to affect FAM122A activity by changing localization and association to 14-3-3 proteins (ref 15), which is not necessarily detected in this setup. FAM122A is clearly phosphorylated in mitosis and there seems to be a difference in absence of greatwall (fig 2C). This should be discussed and some conclusions toned down (e.g. line 132 "These data fully rule out a role of it's phosphorylation", similar reasoning present at more places).

There are hard statements in the text on mitotic entry being abolished after FAM122A depletion, but if I understand correctly, this is all based on one exp showing that centrosomes stay separated longer. I believe that being a central point, further experiments are needed, such as quantifying percentage of cells entering mitosis over time.

Other points

The figure on centrosome separation needs to be clarified. It is stated in the text that the experiment took 40 minutes, but there are data points for more than 40 min. Also, there are averages and SD calculated, but in the text, it is mentioned that cells did not enter mitosis. How are these included to get the average?

Are the point mutants made predicted by alpha-fold to keep the structure of helices?

Point by point response to referee comments

REFERRE #1

1-Point 1:

"Figure 1A/C: It is obvious that the added FAM122A was phosphorylated in the extracts with kinase activity, but the authors did not discuss this or the possible importance of these phosphorylation in FAM122A function. There are a number of S/T.P sites in FAM122A. Is FAM122A a substrate for PP2A-B55"

We agree on the fact that FAM122A is phosphorylated in the extracts containing kinase activity, however, the phosphorylation, although it exerts a positive role, it is not essential for PP2A-B55 inhibition and mitotic entry at doses close to the endogenous ones (respect to calculated FAM122A endogenous concentration in RPE1 human cells). However, we cannot exclude an additional role of this phosphorylation in other regulatory mechanisms such as the binding to 14.3.3 protein. The putative role of FAM122A phosphorylation has been discussed in the results section "lines 280-282".

Point 2:

"Figure 1B: How does level of FAM122A protein added to the Xenopus extract relate to the level of endogenous Xenopus FAM122A?"

We have now measured the minimal dose of wildtype FAM122A able to promote mitotic entry when supplemented to interphase extracts to be between 3 and 15 nM.

Concerning the endogenous concentration of FAM122A in Xenopus extracts, we constructed three different antibodies anti-FAM122A, against the total protein and the Cterminal and Nterminal parts. They recognize the ectopic proteins, but not the endogenous one in the extracts. We additionally performed mass spectrometry to quantify endogenous FAM122A but we could not detect this protein, although we know that it is present because we were able to detect it when we enriched it by purifying phosphopeptides for phosphoproteomics. Since FAM122A is a nuclear protein ¹, we think that we likely lose this protein by centrifugation when purifying Xenopus egg extracts. However, as an alternative, we estimated the levels of endogenous FAM122A in human RPE1 cells to be around 59 nM. This endogenous concentration is higher than the minimal one that promoted mitotic entry in Xenopus egg

extracts (between 3 and 15 nM) suggesting that our phenotype is likely induced under physiological doses of FAM122A although we cannot exclude the fact that the levels of this protein in *Xenopus* could be lower than in human cells.

Point 3:

"Can the authors use BLI or some other biophysical assay to measure the actual affinity of FAM122A binding to PP2A-B55? Does a S62A mutant ARPP19 not have any affinity for PP2A-B55 - in their structural model in Figure 6 there are other protein-protein contacts between ARPP19 and PP2A in addition to the phosphate interactions."

Unfortunately, we have not pure PP2A-B55 complex available to perform biophysical assays. Concerning S62A mutant, we know that this mutant can bind PP2A-B55 however, we need around 3000x more of S62A-Arpp19 than the wildtype protein to detect this binding.

Point 4:

"Figure 2A: Since the blot in the sixth panel was done with anti-pY15 Cdk1 antibodies, I recommend that all the panels of this sort in the paper be labeled anti-pY15 rather than P_{tyr}."

The modifications have been done

Point 5:

"Figure 2F: Again, some comment on the heavy phosphorylation of FAM122A, especially when cyclin A was added in the right-hand panel would be helpful. How many phosphosites are there in FAM122A, and could any of them be relevant to FAM122A/PP2A-B55 interaction?"

This question has been answered in Point 1

Point 6:

" Figure 3A: A linear schematic of the FAM122A protein might also be helpful"

A human and *Xenopus* FAM122A sequence alignment was added to Figure 3A.

Point 7:

" Figure 3E: What is the lower FAM122A band in the WT lane - is it a cleavage fragment? What¹ antibody was used for blotting in this panel?"

The fragment corresponds to a cleavage fragment. The antibody used is anti-Histidine.

Point 8:

" Figure 3F: These mutation data look convincing - can the essential helical residues be displayed on a helical wheel? Can binding of an isolated H1 peptide to B55 be detected? Likewise, does an H2 peptide alone bind/inhibit PP2Ac"

Alanine is prone to helicity thus the alanine mutants will keep alpha helices. The impact of asparagine is much less evident, a deeper analysis would be required.

Concerning the binding of H1 and H2 to PP2A-B55, we checked whether the addition of these peptides could promote mitotic entry. Any of them promoted the entry of interphase extracts in mitosis as shown in the Figure below.

Point 9:

" Figure 5: Different and very precise amounts of Xe FAM122, Thio-Arpp19, His-Arpp19, and GST-Xe FAM122A proteins were added in these experiments, but the rationale for using these amounts was not explained, and it is unclear how the of the final concentrations

achieved by addition of these recombinant proteins related to the endogenous levels of ARPP19 and FAM122A."

In both, Figure 5a and Figure 5b we use a large excess of either Thio-Arpp19 or FAM122A on purpose to determine if we were able to displace FAM122A and Arpp19 respectively bound to PP2A-B55. However, in this new version we performed dose-response assays to identify the minimal dose able to promote entry into mitosis and we compared this dose to the endogenous FAM122A concentration estimated in RPE1 cells. The data confirm that FAM122A can induce mitotic entry at much lower doses than the endogenous one estimated in human cells.

Point 10:

"Figure 5B: The authors should note that GST-FAM122A is dimeric and this will have a significantly higher avidity for PP2A-B55 than monomeric FAM122A, which might affect the interpretation of these experiments."

We agree that GST-FAM122A dimerizes and that this will increase the concentration of FAM122A bound to PP2A-B55, however, in this experiment, even by increasing FAM122A binding to PP2A-B55 by dimerization we do not observe a displacement of Arpp19 from the phosphatase. So, the fact that even upon dimerization GST-FAM122A cannot displace Arpp19 from PP2A-B55 reinforces our hypothesis that once phosphorylated Arpp19 will substitute FAM122A to form an Arpp19-phosphatase complex.

Point 11:

" Figure 5C: This is not described in the text very well, i.e. the fact that this depends upon dephosphorylation of pS67 was not made explicit in the relevant paragraph on page 9."

This point has been clarified in the text lines 250 to 257.

Point 12:

" Figure 6B: The pSer62 (human) and pSerS71 (Xenopus) and pSer61 (C. elegans) residue should be labeled pS62, pS71, pS61 and the phosphate in this residue should be depicted in a different color than amber so it stands out for the reader. Obviously, it would be nice to have

real crystal structures of these complexes to confirm the AlfaFold2-multimer predictions, but this would be beyond the scope of this paper."

The figure has been modified.

REFERRE #2

Point 1:

" Unless the authors have evidence or other justification for the existence of changes in FAM122A activity at the G2-M transition, the title of the paper needs to be changed, and less emphasis placed in the Introduction and Discussion about it being part of a trigger. What remains is still important and justifies publication: FAM122A establishes a precondition needed for cyclinA/cdk to initiate M phase by inhibiting some significant (but currently unknown) fraction of PP2A-B55."

Our new data states that FAM122A levels increase during G2 reaching a maximum at G2-M transition. We have included this data in Figure 8d. However, we have modified the title and the discussion to down modulate the statements about the role of FAM122A as the first event triggering mitosis.

Point 2:

"The authors should at a minimum clarify that the Mn²⁺ PP2A structure is of artifactual, bacterially produced enzyme, that the native enzyme has other divalent cations, and that this fact could significantly alter the way in which pArpp19 interacts with the active site."

We specified this issue in lanes 323 to 326 and lanes 515 to 519.

Point 3:

" The figure legends should be redone with statements of reagent molarities (rather than amounts or volumes) where appropriate. It is often impossible to follow what they did with various dilutions and mixings, and readers should not be forced to calculate molarities themselves."

This point has been addressed in the text.

Point 4:

" It was annoying that the authors rewrote history (in line 298 and 461) that the study establishing pArpp19/Endosulfine works as an inhibitor of PP2A-B55 that is slowly dephosphorylated was their reference published in 2021, when this principle was already established in 2014 (their reference 21)."

We corrected this issue in the text.

Point 5:

" The paper could use some careful editing. Much of the paper is well-written, but here and there sentences crop up with incorrect English constructions that are confusing. I noticed at least one bad typo, in which the concentration of endogenous Arpp19 is stated in line 255 to be 0.115 microM, whereas in Supplementary Figure 2 it is 115 microM. The manuscript is so dense that I gave up trying to document these issues."

This point has been addressed.

Point 6:

"As outlined above, several approaches exist that could clarify the concentration of FAM122A and also the K_i . Some of these approaches are in theory fairly simple: for example, if the authors have antibody that detects FAM122A on Western blots (even if it does not work for immunodepletion), it should be fairly simple to determine the endogenous amounts. Perhaps I am in the minority by being interested in these numbers, but I believe it would increase our understanding of the role of this protein to get some idea of what fraction of PP2A-B55 it is likely to inhibit."

In this new version of our manuscript we performed dephosphorylation assays with kinase-inactivated *Xenopus* egg extracts to identify the K_m of p-S113-Arpp19 and the K_i values of p-S71-Arpp19 and FAM122A. This allow us to stablish that p-S71-Arpp19 displays more than 100-fold higher K_i than FAM122A and to confirm that, by its high affinity, phosphorylated Arpp19 can dissociate FAM122A bound to PP2A-B55.

Point 7:

" It is not clear to me if AlphaFold is sufficiently accurate and robust so that substitution of the divalent cations in the structure of the complex with pArpp19 would show anything not visible in the current model. I also don't know how much computational effort would be involved in doing this substitution. What I do know is that the right divalent cations in the active site do in fact make a huge difference."

AlphaFold does not permit to measure the impact of the substitution of the divalent cations in the structure.

REFERRE #3

Point 1:

"First, it is unclear to me how much FAM122A is added to extracts in relation to the endogenous protein and whether the setup could detect modification of activity. Key experiments in Figure 2 should be reperformed using a titration to test whether the read-out is dose dependent and to assess whether the S/T mutant can affect activity."

These concerns have been addressed in Point 1 and Point 2 of REFEREE#1 answer. See Figure 5E and results section "Lines 266-285.

Point 2:

"Second, phosphorylation of FAM122A has been suggested to affect FAM122A activity by changing localization and association to 14-3-3 proteins (ref 15), which is not necessarily detected in this setup. FAM122A is clearly phosphorylated in mitosis and there seems to be a difference in absence of greatwall (fig 2C). This should be discussed and some conclusions toned down (e.g. line 132 "These data fully rule out a role of it's phosphorylation", similar reasoning present at more places)."

We specified in the text that, although phosphorylation have a positive effect in the inhibitory activity of FAM122A, it is not essential for PP2A-B55 association and cyclin B/cdk1 activation at physiological doses (lanes 278-289).

Point 3:

" There are hard statements in the text on mitotic entry being abolished after FAM122A depletion, but if I understand correctly, this is all based on one exp showing that centrosomes stay separated longer. I believe that being a central point, further experiments are needed, such as quantifying percentage of cells entering mitosis over time."

Data in Figure 8 corresponds to more than 10 different films. In this revised version of the manuscript we measured the number of cells entering mitosis per minute and per gonad. We added these new data in Figure 8c.

Point 4:

" The figure on centrosome separation needs to be clarified. It is stated in the text that the experiment took 40 minutes, but there are data points for more than 40 min. Also, there are averages and SD calculated, but in the text, it is mentioned that cells did not enter mitosis. How are these included to get the average?"

We apologize for this error. Data on Figure 8 have been obtained for more than 10 different timelapse experiments, some of them finished at 40 min, some others at 50 min. We will clarify this issue in the text.

Point 5:

" Are the point mutants made predicted by alpha-fold to keep the structure of helices?"

As reported in Point 8, Referee 1, alanine is prone to helicity thus, alanine mutants will not destabilize the helices. For asparagine, it is not clear whether the mutant could perturb in part helicity.

Dr. Thierry Lorca
CNRS
Cell Cycle
1919 route de Mende
Montpellier 34293
France

22nd Jan 2024

Re: EMBOJ-2023-114793R
Cyclin A/cdk activity and FAM122A-dependent inhibition of PP2A-B55 are key events to trigger mitosis

Dear Anna and Thierry,

Thank you again for submitting your revised manuscript to our editorial office. We have now received the below-copied re-reviews from two of the original referees, in light of which we shall be happy to proceed further with publication in The EMBO Journal. As you will see, both referees do still raise a number of discussion/presentation questions, which I would invite you to answer in a final version of the manuscript, as well as in another dedicated point-by-point response.

In addition, there are a number of editorial issues that need to be addressed at this stage:

- Please upload each main figure file separately without any legend text included; the legends for the main figures should solely be included at the end of the manuscript text file.
- Please remove legends for "supplementary" figures from the main text. Please make sure to rename them (and their in-text references) completely into "Appendix Figure S1/2/3...". The PDF containing them needs to be renamed into "Appendix" and headed by a brief Table Of Contents, which also references their specific page numbers.
- The text contains references to "supplementary tables" that appear to be missing. Are these callouts maybe meant for the "Key Resources Table"? Please correct.
- On the abstract page of the manuscript, please include 4-5 general keyword terms to enhance searchability.
- Please include a dedicated "Disclosure and Competing Interests Statement", in accordance with our updated Guide to Authors (<https://www.embopress.org/competing-interests>)
- As we are switching from a free-text author contribution statement towards a more formal statement based on Contributor Role Taxonomy (CRediT) terms, please remove the present Author Contribution section and instead specify each author's contribution(s) directly in the Author Information page of our submission system during upload of the final manuscript. See <https://casrai.org/credit/> for more information.
- Please adjust the format of the reference list and of the in-text citations according to EMBO Journal format (alphabetical order, author name et al + year.../up to 10 author names in the reference list before et al / please refer to our Guide to Authors for additional information on EMBO J reference format). Also, please adjust the format for citation of preprints as specified in our author guidelines. The citation in the text should be: "(preprint: NAME1 et al, YEAR)"; in the reference list: "Author NAME1, Author NAME2, ... (YEAR) article title. bioRxiv doi: XXX". In this light, please also change the in-text reference for the AlphaFold multimer model to a proper citation with author names and inclusion in the reference list.
- Please double-check to make sure to all relevant funding information in the manuscript is congruent with the info entered into our submission system. Some grants are currently only mentioned in the text but not in the system: ANR-22-0022; La Ligue Nationale Contre le Cancer (Equipe Labellisée, EL2019 CASTRO and EL2018 PINTARD), Ligue Nationale Contre le Cancer (Comité Département 66/ LACROIX); NIH Office of Research Infrastructure Programs (P40 OD010440)
- Please enter valid email addresses for all coauthors in the submission system, so that they could be informed about the submission and final decision. At resubmission, acknowledgement emails failed to be delivered to L. Pintard and JC Labbe.
- Please upload the Source Data file folders, which are currently all combined in a single ZIP archive, separately for each main figure (one zipped folder per main figure); while the Source Data for all Appendix figures should still be uploaded as one single ZIP file. Furthermore, please include the numerical source data (spreadsheets) underlying figures 7C-E and 8A-C
- Please alter the presentation of the data plot in Figure 8D: since N=2, it is not appropriate to plot the average + error bars;

instead, the 2 individual data points need to be plotted for each of the time points.

- Finally, please provide suggestions for a short 'blurb' text prefacing and summing up the conceptual aspect of the study in two sentences (max. 250 characters), followed by 3-5 one-sentence 'bullet points' with brief factual statements of key results of the paper; they will form the basis of an editor-written 'Synopsis' accompanying the online version of the article. Please also upload a synopsis image, which can be used as a "visual title" for the synopsis section of your paper. The image should be in PNG or JPG format, and please make sure that it remains in the modest dimensions of (exactly) 550 pixels wide and 300-600 pixels high.

I am therefore returning the manuscript to you for a final round of minor revision, to allow you to make these modification and upload all modified files; please do not hesitate to contact me if you should want to discuss how to best clarify the referees' remaining queries.

Once we will have received the final re-revised manuscript, we should hopefully be ready to swiftly proceed with formal acceptance and production of the manuscript.

With kind regards,

Hartmut

9) Digital image enhancement is acceptable practice, as long as it accurately represents the original data and conforms to community standards. If a figure has been subjected to significant electronic manipulation, this must be clearly noted in the figure legend and/or the 'Materials and Methods' section. The editors reserve the right to request original versions of figures and the original images that were used to assemble the figure. Finally, we generally encourage uploading of numerical as well as gel/blot image source data; for details see: embopress.org/page/journal/14602075/authorguide#sourcedata

At EMBO Press, we ask authors to provide source data for the main manuscript figures. Our source data coordinator will contact you to discuss which figure panels we would need source data for and will also provide you with helpful tips on how to upload and organize the files.

Further information is available in our Guide For Authors:

In the interest of ensuring the conceptual advance provided by the work, we recommend submitting a revision within 3 months (21st Apr 2024). Please discuss the revision progress ahead of this time with the editor if you require more time to complete the revisions. Use the link below to submit your revision:

Link Not Available

Referee #1:

The authors have made reasonable responses to the points raised by the reviewers. In particular, they have added new data in Figure 5E to address issues raised about the relative affinities of FAM122A and phosphoARPP19 for PP2A-B55 and in Figure 8 to address the question about endogenous FAM122A protein levels.

1. The new data in Figure 5E show that the nonphosphorylatable mutant FAM122A is 5-fold less effective at inducing entry into mitosis than the WT FAM122A, implying the mitotic phosphates on FAM122A are functionally significant. However, it should be noted that mutating all the Ser/Thr residues in FAM122A to Ala may have harmful effects on FAM122A function that are independent of the fact that Ser/Thr phosphorylation of FAM122A is no longer possible, i.e., loss of certain OH groups per se may be deleterious. Despite this experiment, it remains unclear how the mitotic phosphates on FAM122A enhance its activity as a PP2A-B55 inhibitor. Admittedly, however, an exhaustive mutational analysis of which Ser/Thr residues are important for FAM122A mitotic function and whether these residues are in fact phosphorylated would be beyond the scope of this paper.

2. The new data in Figure 8 are helpful, especially the demonstration that FAM122A protein levels increase during G2 and then fall after exit from M in the RPE1 human cell line. However, it is curious that the authors were unable to detect any endogenous FAM122A in the *Xenopus* egg extract - they argue that this is because FAM122A is normally nuclear and therefore lost when egg extracts are prepared, but in general amphibian eggs have large stockpiles of nuclear proteins that are required to sustain several rounds of cell division in the absence of protein synthesis and they are not all in the nucleus. If FAM122A is normally retained in the nucleus throughout interphase, this raises the question at stage FAM122A acts to promote cyclin B/Cdc2 activation during the entry into mitosis, since cyclin B/Cdc2 is excluded from the nucleus - does FAM122A "escape" during nuclear envelope breakdown in prophase? In this regard, is FAM122A exclusively nuclear in human RPE1 cells? If there are extremely low or zero levels of FAM122A in the egg extract, does this mean that the mitotic cycling observed when DNA is added to the egg extract occurs normally in the absence of FAM122A?

3. The inclusion of a schematic incorporating the authors' final conclusions about FAM122A being supplanted by phosphoARPP19 to provide complete PP2A-B55 phosphatase inhibition might be helpful to the reader.

Referee #2:

In my review of the original submission of this paper, I favored publication of a revised edition in The EMBO Journal. The authors have undertaken a good faith effort to address the concerns of myself and the other reviewers, and I remain in favor of publication. However, I am puzzled by the implications of some of the newer results and commentary particularly related to the experiments in *Xenopus* extracts, and feel that the authors should address at least one of these implications in their Discussion. I do not see the need for further experimentation.

Perhaps the strongest aspect of the manuscript concerns the experiments done in *C. elegans*, which establish the importance of FAM122A for mitotic progression in at least some cell types. I will not comment further on those experiments. The structural

analysis of the binding of FAM122A and ARPP19/ENSA to PP2A-B55 was also a strong part of the original submission. This structural analysis has been somewhat superseded by an article published a few weeks ago reporting the cryo-EM structures of these complexes (Nature 625: 195-203). POINT 1: It would be appropriate for this manuscript to mention, either as a Note Added in Proof or as a few sentences in the text the degree to which their structural analysis is in agreement or disagreement with the conclusions of the article in Nature. (By the way, the Nature article used PP2A-B55 made in eukaryotic cells and thus likely has the correct divalent cations in the active site.)

One of the more puzzling aspects of the original submission is that the authors have been unable to detect FAM122A in the interphase extracts they used, either with several independent antibodies or by mass spectrometry. (Apparently however, when the extracts are enriched for phosphoproteins, they have detected at least one FAM122A-specific phosphosite.) In this revision, the authors added the speculation that FAM122A resides mostly in the nucleus, and the nuclei were lost during the extract preparation. This speculation may be correct, but if so, that raises some questions. First, some *Xenopus* extracts like CSF extracts are made from oocytes in which the nuclear envelope has broken down, and one might expect that the protein would be revealed on Western blots of such extracts. (Or detected simply by crushing oocytes in SDS.) Was this not done? Second, cycling extracts of *Xenopus* oocytes can be made that undergo several rounds of M phase entry and exit. If FAM122A is not present in those extracts in detectable amounts, one might conclude that FAM122A is not required for cell cycle progression in *Xenopus* extracts. (Though this would not preclude its requirement in intact *Xenopus* eggs; see below.) I am not sure what if anything should be done about these concerns and would leave it to the authors' discretion.

A second and perhaps related area of puzzlement concerns the results shown in new Figure 5e, which demonstrated that 15 nM of ectopic FAM122A is sufficient to drive *Xenopus* extracts into M phase. It is somewhat weird that their other extract experiments employ a 500X excess of exogenous FAM122A. However, as most of those experiments are designed to show that Arpp19/ENSA can displace FAM122A from the phosphatase, I am not (much) bothered by these heroic amounts of FAM122A. The displacement results are completely in accordance with, and can be predicted from, the K_i values that the authors have measured in new experiments. These K_i measurements are likely quite accurate as the value for the K_i of thiophospho Arpp19/ENSA for PP2A-B55 is near identical with previously published values using purified components. POINT 2: On this note, the authors should reword lines 529 and 530, which state that FAM122A "has to be" dissociated from PP2A-B55 by phosphoArpp19/ENSA. The authors have demonstrated that this displacement occurs and in a reasonable time course. However, insofar as I can tell they have not demonstrated this this displacement is in fact necessary for cell cycle progression.

The result that 15 nM FAM122A is sufficient to drive M phase is unexpected (at least to me) because any way you look at it, FAM122A could interfere with the activity of only a small proportion of PP2A-B55. (a) The authors have not quantitated the concentration of PP2A-B55 in their extracts, but other estimates of the same extracts have been in the range of 100-300 nM. If the lowest of these estimates is correct, then even if all the ectopic FAM122A is bound to the phosphatase, suppression of only 15% of the enzyme activity is sufficient to drive the extracts into M phase. (This 15% is in addition to whatever other percent is inhibited by endogenous FAM122A, whose concentration we don't know but is likely minimal in the extracts.) (b) The authors have now done some nice experiments that indicate the K_i of FAM122A for PP2A-B55 is about 28 nM. As stated above, this value is likely to be quite accurate. But now, at least in a two-component system containing 100 nM of PP2A-B55, 15 nM of FAM122A would be able to suppress only ~11% of PP2A; if [PP2A-B55] is 200 nM, then this would be 6.5%. (c) All these values are likely large overestimates of the proportion of PP2A-B55 bound by this ectopic amount of FAM122A because in the extract, the phosphate is bound by a plethora of other substrates and possible inhibitors. I went through these calculations in my previous review and won't repeat them, but very roughly, given 15 nM of FAM122A and a K_i of ~28 nM, and a favorable sum of $(\frac{S}{K_m} + \frac{I}{K_i})$ calculated from Mochida's dilution experiments, you would expect that the activity of PP2A-B55 in the extract supplemented with ectopic FAM122A would be ~98% of the activity without it. Yet this small change is apparently sufficient to drive the extract into M phase.

I'm unsure how the authors should approach this. Points (a) and (b) above are based on the known concentration of ectopic FAM122A and careful measurements of the K_i . It is possible that previous measurements of [PP2A-B55] are too high; for example, it is hard to determine if those estimates were based on the concentration in the extract or were extrapolated to the concentration in the oocyte (~2-3X higher). But in any case, I do not think the authors should be required to reinvestigate this point unless they themselves are motivated to do so. Point (c) is my own way of looking at the situation which the authors may not share and in any case is difficult to explain.

POINT 3: Perhaps the best solution would be to add a paragraph saying that it is remarkable that such a low concentration of FAM122A is sufficient to drive M phase because it is likely to affect only a small proportion of PP2A-B55. This fact indicates that the system is very delicately balanced, so that a small change in PP2A-B55 activity has a strong nonlinear effect. It is also possible that FAM122A is not uniformly distributed in the extract: perhaps it accumulates in the vicinity of key Cdk/CyclinA-directed phosphosites that become PP2A-B55 substrates. In this regard, it is surprising that the authors do not make a larger point of considering the regulated intranuclear location of FAM122A. In cells, movement of FAM122A into nuclei late in G2 might play a larger role in driving M phase than the relatively modest increase in FAM122A concentration noted by the authors; after all, Cdk/CyclinA and its key substrates are located there.

Dear Editor,

Enclosed you will find that revised version of our manuscript entitled "The increase of Cyclin A/cdk activity and of FAM122A-dependent inhibition of PP2A-B55 are key events to trigger mitosis" by Lacroix et al. (EMBOJ-2023-114793) that address all the concerns of the two referees.

REFEREE #1

Point 1:

The new data in Figure 5E show that the nonphosphorylatable mutant FAM122A is 5-fold less effective at inducing entry into mitosis than the WT FAM122A, implying the mitotic phosphates on FAM122A are functionally significant. However, it should be noted that mutating all the Ser/Thr residues in FAM122A to Ala may have harmful effects on FAM122A function that are independent of the fact that Ser/Thr phosphorylation of FAM122A is no longer possible, i.e., loss of certain OH groups per se may be deleterious. Despite this experiment, it remains unclear how the mitotic phosphates on FAM122A enhance its activity as a PP2A-B55 inhibitor. Admittedly, however, an exhaustive mutational analysis of which Ser/Thr residues are important for FAM122A mitotic function and whether these residues are in fact phosphorylated would be beyond the scope of this paper.

We agree with the referee on this point, but, in this case, the mutation of all the Sr/Thr to alanine did not prevent the capacity of FAM122A to inhibit PP2A-B55. However, experiment in Figure 5E suggests that these phosphorylations would have a positive modulation of the inhibitory activity of this protein. We pointed out this issue in the discussion of the new version of the manuscript (lines 482-484).

Point 2:

The new data in Figure 8 are helpful, especially the demonstration that FAM1122A protein levels increase during G2 and then fall after exit from M in the RPE1 human cell line. However, it is curious that the authors were unable to detect any endogenous FAM122A in the *Xenopus* egg extract - they argue that this is because FAM122A is normally nuclear and therefore lost when egg extracts are prepared, but in general amphibian eggs have large stockpiles of nuclear proteins that are required to sustain several rounds of cell division in the absence of protein synthesis and they are not all in the nucleus. If FAM122A is normally retained in the nucleus throughout interphase, this raises the question at stage FAM122A acts to promote cyclin B/Cdc2 activation during the entry into mitosis, since cyclin B/Cdd2 is

excluded from the nucleus - does FAM122A "escape" during nuclear envelope breakdown in prophase? In this regard, is FAM122A exclusively nuclear in human RPE1 cells? If there are extremely low or zero levels of FAM122A in the egg extract, does this mean that the mitotic cycling observed when DNA is added to the egg extract occurs normally in the absence of FAM122A?

Concerning the activation of cyclin B/Cdk1 by FAM122A, we didn't check the nuclear localization of FAM122A at G2-M transition in RPE1 cells, however, we could observe that the germinal cells from *C. elegans* did actually present a nuclear localization of FAM122A during interphase but this protein rapidly exit the nucleus just before nuclear envelope breakdown suggesting that this cytoplasmic localization will participate to the activation of cyclin B/cdk1 in the cytosol. This question is currently being addressed in our laboratory.

Concerning the capacity of cycling extracts to perform several cell cycles, this is mostly controlled by mRNA translation. Most, cell cycle proteins including cyclin B, cyclin A, c-mos ... are stocked in the *Xenopus* oocytes as mRNAs whose translation is activated by meiotic resumption upon progesterone addition. Mitotic cycling induced by DNA sperm will be accompanied by the translation of the different cell cycle mRNAs inducing the synthesis of cyclin B, cyclin A... and probably also FAM122A that will be then relocalized in the nuclei formed upon DNA sperm addition.

Point 3:

The inclusion of a schematic incorporating the authors' final conclusions about FAM122A being supplanted by phospho ARPP19 to provide complete PP2A-B55 phosphatase inhibition might be helpful to the reader.

A scheme has been added.

REFEREE #2

Point 1:

Perhaps the strongest aspect of the manuscript concerns the experiments done in *C. elegans*, which establish the importance of FAM122A for mitotic progression in at least some cell types. I will not comment further on those experiments. The structural analysis of the binding of FAM122A and ARPP19/ENSA to PP2A-B55 was also a strong part of the original

submission. This structural analysis has been somewhat superseded by an article published a few weeks ago reporting the cryo-EM structures of these complexes (Nature 625: 195-203). It would be appropriate for this manuscript to mention, either as a Note Added in Proof or as a few sentences in the text the degree to which their structural analysis is in agreement or disagreement with the conclusions of the article in Nature. (By the way, the Nature article used PP2A-B55 made in eukaryotic cells and thus likely has the correct divalent cations in the active site.)

The manuscript and the agreement and disagreement points of our study with this manuscript has been added in the discussion of the revised version of the manuscript (lines 537-552).

Point 2:

One of the more puzzling aspects of the original submission is that the authors have been unable to detect FAM122A in the interphase extracts they used, either with several independent antibodies or by mass spectrometry. (Apparently however, when the extracts are enriched for phosphoproteins, they have detected at least one FAM122A-specific phosphosite.) In this revision, the authors added the speculation that FAM122A resides mostly in the nucleus, and the nuclei were lost during the extract preparation. This speculation may be correct, but if so, that raises some questions. First, some *Xenopus* extracts like CSF extracts are made from oocytes in which the nuclear envelope has broken down, and one might expect that the protein would be revealed on Western blots of such extracts. (Or detected simply by crushing oocytes in SDS.) Was this not done? Second, cycling extracts of *Xenopus* oocytes can be made that undergo several rounds of M phase entry and exit. If FAM122A is not present in those extracts in detectable amounts, one might conclude that FAM122A is not required for cell cycle progression in *Xenopus* extracts. (Though this would not preclude its requirement in intact *Xenopus* eggs; see below.) I am not sure what if anything should be done about these concerns and would leave it to the authors' discretion.

We checked the levels of FAM122A in CSF extracts but, as for interphase extracts, we could not detect this protein. However, our data on RPE1 indicates that FAM122A levels follow those of cyclin A that degrades in prometaphase. Thus, it is possible that we cannot detect this protein in CSF extracts because of its degradation.

As pointed out in point 2 of Referee 1, it is possible that FAM122A mRNA could be translated as for cyclin A or cyclin B during the first division of the *Xenopus* oocytes thus providing the level of FAM122A required for recycling. This is a very interesting point that we will address in future experiments by inducing maturation of *Xenopus* oocytes.

Point 3:

On this note, the authors should reword lines 529 and 530, which state that FAM122A "has to be" dissociated from PP2A-B55 by phosphoArpp19/ENSA. The authors have demonstrated that this displacement occurs and in a reasonable time course. However, insofar as I can tell they have not demonstrated this displacement is in fact necessary for cell cycle progression.

The sentence: FAM122A "has to be" dissociated from PP2A-B55 by phosphoArpp19/ENSA has been modified (lines 553-555).

Point 4:

Perhaps the best solution would be to add a paragraph saying that it is remarkable that such a low concentration of FAM122A is sufficient to drive M phase because it is likely to affect only a small proportion of PP2A-B55. This fact indicates that the system is very delicately balanced, so that a small change in PP2A-B55 activity has a strong nonlinear effect. It is also possible that FAM122A is not uniformly distributed in the extract: perhaps it accumulates in the vicinity of key Cdk/CyclinA-directed phosphosites that become PP2A-B55 substrates. In this regard, it is surprising that the authors do not make a larger point of considering the regulated intranuclear location of FAM122A. In cells, movement of FAM122A into nuclei late in G2 might play a larger role in driving M phase than the relatively modest increase in FAM122A concentration noted by the authors; after all, Cdk/CyclinA and its key substrates are located there.

We added a text in the discussion of the revised version of the manuscript (lines 496-502) a paragraph pointing out that the lower concentration of FAM122A able to promote mitotic entry would promote the inhibition of a very small proportion of PP2A-B55. We also included a sentence proposing a putative role of the specific localization of FAM122A at the vicinity of cyclin A/cdk1. Concerning the subcellular localization of FAM122A, as exposed in point 2 of referee 1, we are currently investigating the putative role of this mechanism in G2-M transition since we observed a translocation of FAM122A prior nuclear envelope breakdown in *C elegans* germinal cells.

Hoping for a positive answer

Sincerely

Dr. Thierry Lorca
CNRS
Cell Cycle
1919 route de Mende
Montpellier 34293
France

6th Feb 2024

Re: EMBOJ-2023-114793R1
Increases in cyclin A/Cdk activity and in PP2A-B55 inhibition by FAM122A are key mitosis-inducing events

Dear Thierry and Anna,

Thank you for submitting your final revised manuscript for our consideration. I am pleased to inform you that we have now accepted it for publication in The EMBO Journal.

With kind regards,

Hartmut
